# iQR: Quantile Regression with QR Orthogonal Decomposition for Resource Scheduling Optimization without Empirical Model

## Abstract

Optimal resource scheduling aims to cover resource demand with minimum economic cost, which is far from model-based constraint optimization or data-driven prediction based scheduling process. To address this model-free constraint optimization issue, a sparse system identification framework with quantile regression and QR orthogonal decomposition (iQR) is proposed for complex systems ranging from small to large scales. It leverages quantile optimization with $L_1$-norm to construct a business-driven strategy to reach the proportion of meeting resource demand. It also involves a complete-mapping Fourier Transformation process and an orthogonal least squares technique to select basis vectors in advance to achieve fast regression with sparse mathematical expression, which reduces the number of basis functions from thousands to hundreds and to dozens. iQR represents a specific expectation for single time series prediction, which only achieves predictions that deviate from the true values as little as possible, but aims for predictions consistently higher than the actual values of real demand. Numerical experiments was conducted on eight datasets, including commonly used time series and real-world CPU resource data. The results indicate that most neural network-based methods fail to balance both resource demands and prediction accuracy effectively. In contrast, iQR can achieve optimal scheduling with the minimum economic cost and it is easier to satisfy the business constraints with quantile tuning. Notably, iQR is lightweight with a training speed in seconds and does not rely on the support of computing power of GPU resources. This study may provide new insight into investigations on resource scheduling optimization issues.

## 1 Introduction

Optimal resource scheduling seeks to meet demand with minimal cost, which is essential across various industries such as services, energy, and communications due to its substantial economic benefits (He et al., 2012; Zhang et al., 2018). Such constraint optimization problems often lack explicit formulas and is typically solved by either model-based planning methods or data-driven prediction based approaches. However, both of them have significant shortcomings with little consideration of empirical modeling errors or business constraints. Recently, numerous methods have been proposed to tackle this issue. Firstly, model-based optimization techniques are applied in resource scheduling, such as NFS (Jones & Peet, 2021), STS (Wang et al., 2021), and DD-TMPC (Zhao et al., 2024). Despite their utility, these methods struggle to capture the complexities of dynamic environments and incur significant computational costs for model maintenance and updates. Secondly, prediction-based scheduling solutions are also prevalent. Contemporary neural network approaches for time series forecasting include models like iTransformer (Liu et al., 2023), FITS (Xu et al., 2024), PatchTST (Nie et al., 2023), and TimesNet (Wu et al., 2023). These models are capable of providing long-term forecasts for a variety of multivariate data types, including electricity, exchange rates, weather, and disease data (Zhou et al., 2022; Zhang & Yan, 2022; Zeng et al., 2023; Bi et al., 2023; Zhou et al., 2024). However, these methods fall short of adequately meeting business requirements, even when the objective function is simply modified to quantile regression to provide a confidence of forecasting. Additionally, they necessitate GPU resources, which are not accessible to some smaller laboratories. Thirdly, direct application of quantile regression for scheduling is another approach. These methods

provide good interpretability and aligns well with business requirements (Zarnani et al., 2019; Zhang et al., 2018). Nonetheless, due to the inherent complexity of time series data, these methods often require extensive running time, which exceeds practical tolerance limits. In light of these challenges, we aim to develop a lightweight, business-oriented, more interpretable (with explicit formulas) framework for resource scheduling problems. This framework will be specifically designed to meet business requirements and deliver accurate and rapid predictions without relying on complex neural network models or extensive pre-training process.

Thus, we propose a sparse system identification framework with quantile regression and QR orthogonal decomposition (iQR) for optimal resource scheduling. Specifically, iQR applies the Fourier transform to time series, selecting the most influential basis vectors to build a basis vectors library. Additionally, we apply Seasonal-Trend decomposition using LOESS (STL) decomposition (Cleveland et al., 1990) to the input sequence, allowing individual identification for different time series and improving the precision of long-term scheduling. To enhance efficiency, we initially screen the basis vectors before incorporating business constraints. We design the Stepwise Orthogonal Regression (SOR) algorithm to retain only the most important basis vectors for training. Subsequently, we employ $L_1$-norm optimization with quantile regression to solve the business-driven problem. The main contributions of iQR can be summarized as follows:

- A novel business-driven identification-prediction framework has been introduced, offering remarkable flexibility by enabling the customization of the objective function according to specific business satisfaction levels. It enables rapid, cost-effective scheduling across various scenarios, offering new insights for future resource scheduling applications.

- By incorporating global information in the form of Fourier bases into local identifications and leveraging STL decomposition to differentiate identifications, iQR enables long-term and comprehensive identifications for time series. This innovative approach combines data-driven techniques with explicit mathematical expressions, enhancing both interpretability and scheduling capabilities.

- The specially designed SOR algorithm for the further selection of basis vectors and the utilization of quantile optimization accelerate the optimization process with sparse mathematical expression, which reduces the number of basis functions from thousands to hundreds and even to dozens. This contributes to the investigation of regression acceleration, and enables the iQR framework to be swiftly applied in business-driven scheduling scenarios.

- Impressive scheduling accuracy was achieved on eight benchmark datasets, surpassing current state-of-the-art (SOTA) outcomes. Our method is more effective in satisfying scheduling demands, and when constraints are met, it incurs significantly lower economic costs. Notably, iQR is lightweight and does not rely on GPU computing power support during the training process compared with neural network based methods.

## 2 RELATED WORK

**Model-based Optimization for Scheduling**  Model-based scheduling optimization methods utilize explicit empirical models of the environment to optimize schedules. Based on the well-designed model, many works have been proposed to address the resource scheduling issues with convex or non-convex optimization solving approaches, including NFS (Jones & Peet, 2021), STS (Wang et al., 2021), DD-TMPC (Zhao et al., 2024). However, several drawbacks exist in these methods, e.g., the difficulty of accurately modelling complex dynamics and the high computational cost on updating models. Model-free methods are increasingly gaining attention due to their enhanced adaptability and capability of handling complex dynamics behind the data. Latest works develop the model-free scheduling policy for optimization, involving Zhao et al. (2023); Zhang et al. (2024). Model-free optimization strategies directly learn optimal policies from interactions without explicit environmental models, and are more robust to changes in the dynamical process.

**Prediction-based Scheduling**  Recently, methods for time series prediction using neural networks have gained widespread attention. Transformer-based methods aim to continuously reduce the time complexity of the model while improving accuracy. These methods include Timemixer (Wang et al., 2024), TimesNet (Wu et al., 2023), Basisformer (Ni et al., 2023), iTansformer (Liu et al., 2023), PatchTST (Nie et al., 2023), FEDformer (Zhou et al., 2022), FITS (Xu et al., 2024), and others.

Despite the significant advancements of Transformer-based methods, their accuracy and training speed can still be unsatisfactory in some cases. As a result, two optimization approaches have emerged. The first approach involves leveraging larger models (Zhou et al., 2024; Cao et al., 2024) to further enhance accuracy, but this leads to slower total times. The second approach involves abandoning the Transformer-based framework and instead implementing plain neural networks (Zeng et al., 2023; Xu et al., 2024; Das et al., 2023) for time series prediction. Unfortunately, these deep learning methods often excel at solving a specific tasks, and their performance may decline when prediction requirements change or business constraints are introduced.

**Quantile Regression-based scheduling** Common quantile regression include FLM (Zarnani et al., 2019), NWPs (Zarnani et al., 2019) or traditional model TRMF (Yu et al., 2016), SINDy (Brunton et al., 2016), TSM (Gao & Yan, 2022) with quantile regression. These models can achieve quantile regression by utilizing their own basis function library without relying on the backpropagation process of the support of GPU. Moreover, there are existing methods that can handle business-driven problems (Dai, 2023; Wang & He, 2024; Li & Zhu, 2008). However, most of these methods lack open-source code, which limits their accessibility. Compared to data-driven approaches, these methods may face challenges in making long-term scheduling, and the prediction accuracy may not be as high as that of neural network-based methods. Besides, when incorporating business-driven constraints into them, although they can be solved in a CPU environment, the execution speed becomes significantly slower, especially when dealing with large basis function libraries. This poses a challenge in meeting the speed requirements of resource scheduling.

## 3 METHODOLOGY

This section details the framework and modules of iQR. Firstly, the overall structure of iQR is presented in Section 3.1, as well as the roles of the key components if iQR. Then, the two-phase identification process including the global-local coordination mechanism is comprehensively introduced in Section 3.2 and 3.3. Moreover, the theoretical description regarding the key basis vectors selection algorithm, Stepwise Orthogonal Regression (SOR), is presented in Section 3.4.

**Problem Definition** Given the historical sequence representing the resource demand at timestamp $t$, referred to $\mathbf{X}_t = [X_{t-H+1}, X_{t-H+2}, \ldots, X_t] \in \mathbb{R}^{H \times d_x}$ of length $H$ , the task is to generate a future sequence $\hat{\mathbf{Y}}_t = [\hat{Y}_{t+1}, \hat{Y}_{t+2}, \ldots, \hat{Y}_{t+P}]$ of length $P$. Its values exceed the ground truth $\mathbf{Y}_t = [Y_{t+1}, Y_{t+2}, \ldots, Y_{t+P}] \in \mathbb{R}^{P \times d_y}$ while minimizing the gap between them, ensuring efficient resource allocation. Here, $d_x$ and $d_y$ denote the dimensions of input and output features, respectively. The problem is denoted as

$$
\begin{aligned}
\min \quad & \left( \hat{\mathbf{Y}}_t - \mathbf{Y}_t \right) \cdot \mathbb{I} \left( \hat{\mathbf{Y}}_t \geq \mathbf{Y}_t \right), \\
s.t. \quad & \mathbb{I} \left( \hat{\mathbf{Y}}_t \geq \mathbf{Y}_t \right) \geq \tau,
\end{aligned}
\tag{1}
$$

where $\mathbb{I}(u)$ is the indicator function, equals to 1 if $u \geq 0$ and 0 otherwise. The quantile $\tau \in [0, 1]$ represents the percentage of identification results that are greater than or equal to the true values. For further visualization and better understanding, please refer to Appendix B. Note that Eq. (1) established in a resource-constrained environment, e.g., CPU resource scheduling in cloud networks, is the focal issue in this study, which is far from a single prediction problem with no business constraint here. One must first satisfy the constraint, then address the optimization.

### 3.1 iQR FRAMEWORK

iQR is a business-driven framework based on system identification, combining basis selection strategy and quantile regression. As shown in Figure 1, it consists of two main phases: global identification and local identification. The two-phase identification (Liu et al., 2024) process is adopted to capture macroscopic and microscopic trends, respectively. Each channel of iQR contains only one univariate sequence, and each sequence is modeled separately. For channel $i$, the sequence $X^i$ is split into a training set $X_{train}^i$ and a test set $X_{test}^i$. Concretely, the Fast Fourier Transform (FFT) is employed to extract periods $\mathbf{T}$ in the series, which are used to construct the basis vectors library $\boldsymbol{\Theta}$. Stepwise Orthogonal Regression (SOR) eliminates independent basis vectors and yields significant ones $\hat{\boldsymbol{\Theta}}$,

which plays the role of second basis selection. The $L_1$-norm Quantile Regression ($L_1$QR) ensures output $\hat{Y}$ exceeds the ground truth $Y$, controlling the lower bound of output results.

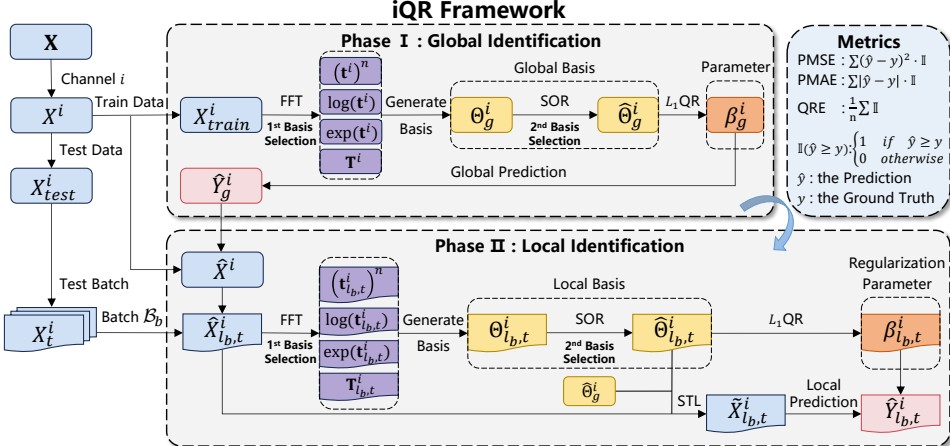

Figure 1: The overall framework of iQR. Global identification captures macroscopic changes. Local identification utilizes macro trends and investigates micro variations to achieve rolling predictions.

The roles of the key components of iQR are described as follows:

**Global-Local Coordination**   The coordination identification strategy aims at dynamic analysis of input series. The global identification focus on the pivotal trends, whereas the local identification leverages the macro information provided by its global counterpart, digs micro sensitive variations.

**FFT**   The Fourier transform is used to reveal the complete description in frequency domain, since mostly-adopted polynomial basis vectors inspired by the Taylor's formula can only denote the spontaneous feature of a single point with even known nonlinear function of the signal.

**SOR**   SOR is the key selection method of basis vectors suited for sequence identification. It selects the influential ones that contributes most to the model, and ensures the selection at each step is independent and valid by orthogonal projection. The basis vectors are then input into $L_1$QR for identification.

$L_1$**QR**   To facilitate sparsity among model coefficients, the LASSO penalty term is introduced into the loss function. $L_1$-norm Quantile Regression effectively shrinks unimportant or irrelevant feature weights to zero, thereby generates a robust, compact, and interpretable model structure.

## 3.2 GLOBAL IDENTIFICATION

In the global identification phase, the primary objective is to extract meaningful patterns and trends from historical data that can serve as the foundation for subsequent local identification.

Concretely, global identification applies FFT to the training data. This mathematical operation transforms the time-domain signal into its frequency-domain representation, revealing the underlying frequency components $f = \text{FFT}(\mathbf{X})$. Then, we identify the most salient frequencies, typically those with the highest amplitudes. Using the frequency-period relationship $f = 1/T$, we convert these significant frequencies into their corresponding periods $\mathbf{T}^i = [T_1^i, T_2^i, \ldots, T_n^i]$. These periods present the characteristic timescales or cyclic behaviors that are influential in shaping the time series dynamics. Based on the representative periods, the global basis vectors is presented as:

$$\mathbf{\Theta}_g^i = \left[ (\mathbf{t}^i)^n, \log(\mathbf{t}^i), \exp(\mathbf{t}^i), \sin\left(\frac{2\pi \mathbf{t}^i}{\mathbf{T}^i}\right), \cos\left(\frac{2\pi \mathbf{t}^i}{\mathbf{T}^i}\right) \right], \tag{2}$$

where $n \in \{0, 1, \ldots\}$, the timestamp set $\mathbf{t}^i$ is $[0, 1, \ldots, L_{train}]$, and $L_{train}$ is the length of the training set.

Rather than employing a fixed set of basis vectors, we utilize SOR to dynamically select the most informative and relevant basis vectors from the initial basis set identified in the previous phase, the basis vectors are $\hat{\Theta}_g^i$, the sparse system identification process with $L_1$QR can be represented as

$$\arg\min_{\beta_g^i} \sum_{i=1}^{n} \rho_\tau \left( Y^i - X^i \beta_g^i \right) + \lambda ||\beta_g^i||_1. \tag{3}$$

Given the coefficients $\beta_g^i$, the global identification is calculated by $\hat{Y}_g^i = X^i \beta_g^i$. Different from methods based on neural networks, the global identification phase utilizes the entire training set for identification, thus reducing the likelihood of accumulated errors propagating into future outputs.

### 3.3 LOCAL IDENTIFICATION

The local identification phase builds upon the knowledge of global identification and focuses on the accurate rolling output on the test set.

Concretely, the overall architecture of local identification is similar to global identification phase, and the pivotal distinction lies in the segmentation of the test dataset into numerous batches, denoted as $\mathcal{B}_b$, for $i$ ranging from 1 to $N$. Conventional regression methods and advanced deep learning frameworks customarily generate subsequent sequence directly from the test sequence. However, the neglect of global trends significantly decrease the prediction accuracy. Consequently, a knowledge fusion technique is introduced, which is accomplished via a weighted mean strategy that synthesizes the global inferred sequence $Y_g^i$ with the local entries $X^i$, formalized as follows:

$$\hat{X}^i = \mathbf{w}X^i + (1 - \mathbf{w})\hat{Y}_g^i, \tag{4}$$

where $\mathbf{w}$ is a vector constructed by generating $L_{batch}$ equally spaced points on the interval $[\alpha^{1/\gamma}, \xi^{1/\gamma}]$, and then raising each of these points to the power of $1/\gamma$. Formally, $w_i = (\alpha^{1/\gamma} + b\Delta)^\gamma$, for $b = 0, \dots, L_{batch} - 1$ and $\Delta = (\xi^{1/\gamma} - \alpha^{1/\gamma})/(L_{batch} - 1)$ defines the uniform spacing between consecutive points. The weighted coefficient $\mathbf{w}$ balances the influence of overarching tendencies against localized fluctuations. Applying the strategy to $\mathcal{B}_b$ yields the refined inputs.

FFT is used to extract an initial set of basis vectors. For a test batch $\mathcal{B}_b$, the local basis vectors are

$$\Theta_{l_b,t}^i = \left[ (\mathbf{t}_{l_b,t}^i)^n, \log(\mathbf{t}_{l_b,t}^i), \exp(\mathbf{t}_{l_b,t}^i), \sin\left(\frac{2\pi\mathbf{t}_{l_b,t}^i}{\mathbf{T}_{l_b,t}^i}\right), \cos\left(\frac{2\pi\mathbf{t}_{l_b,t}^i}{\mathbf{T}_{l_b,t}^i}\right) \right], \tag{5}$$

where $\mathbf{t}_{l_b,t}^i$ uniformly samples from 0 to the length of the training part in test batch $\mathcal{B}_b$ and $\mathbf{T}_{l_b,t}^i$ defines the representative periods. To refine and emphasize higher frequency constituents, the SOR algorithm is invoked, leading to the optimized local basis vectors $\hat{\Theta}_{l_b,t}^i$. Before applying the selected basis vectors for identification, the local time series is enhanced by performing STL decomposition. STL decomposes the data into seasonal, trend, and residual components, enabling isolate and adjust for any seasonality presented in the test data. The resulting adjusted series serves as the input for the next step. After the LASSO identification, the input sequence is

$$\tilde{X}_{l_b,t}^i = \text{LASSO}(\mathcal{S}_{l_b,t}^i) + \text{LASSO}(\mathcal{T}_{l_b,t}^i) + \mathcal{R}_{l_b,t}^i. \tag{6}$$

For the input series $\tilde{X}_{l_b,t}^i$, the objective function of $L_1$QR is

$$\arg\min_{\beta_{l_b,t}^i} \sum_{i=1}^{n} \rho_\tau(Y_{l_b,t}^i - \tilde{X}_{l_b,t}^i \beta_{l_b,t}^i) + \lambda ||\beta_{l_b,t}^i||_1. \tag{7}$$

Utilizing the fitted model, we perform local rolling predictions by iteratively updating the model with new observations. This process continually adapts the model to the most recent data, capturing local fluctuations while maintaining the global context provided by the global identification.

### 3.4 THE KEY MODULE OF iQR FRAMEWORK: SOR

To identify significant basis vectors and further eliminate irrelevant ones, the key module of iQR, Stepwise Orthogonal Regression (SOR), has been proposed. The detail theory is shown as follows.

Given the initial set of basis vectors $\Theta = \{\theta_1, \theta_2, \dots, \theta_p\}$, we employ the following Lemma 1 to ensure orthogonality among basis vectors.

**Lemma 1.** *(QR Decomposition (Golub & Van Loan, 2013)) Given a real $n \times n$ square matrix $A$, there exist $n \times n$ orthogonal matrix $Q$ and upper triangular matrix $R$ such that $A = QR$.*

Then, the Lemma 1 is extended to a general scenario, that is, matrices of arbitrary dimensions.

**Corollary 1.** *Any real $m \times n$ matrix $A$ can be decomposed into the product of an $m \times n$ orthogonal matrix $Q$, whose columns are mutually orthogonal, and an $n \times n$ upper triangular matrix $R$.*

**Corollary 2.** *Any real $m \times n$ matrix $A$ can be factorized into the product $A = VR$, where $V$ is an orthogonal matrix with its columns being pairwise orthogonal, and $R$ is an upper triangular matrix with diagonal entries equal to 1.*

According to Corollary 2, the basis vectors $\Theta$ is orthogonalised to filter the insignificant basis vectors, and sbsequently serve for the later lemma. The following lemmas show a stepwise regression strategy to efficiently select the optimal subset of basis vectors.

**Lemma 2.** *(Orthogonality Principle) For an unbiased optimal estimator (i.e., the least squares estimation), if the basis vectors are orthogonal, then the error is orthogonal to any basis vector.*

**Lemma 3.** *(Orthogonal Greedy Optimality) If the basis vectors are orthogonal, then a greedy stepwise optimal selection of basis vectors will yield the same result as the global optimal solution.*

**Remark 1.** *Since the basis vectors are orthogonal, the contribution of each function to the model can be evaluated independently of the others. At each step of the greedy algorithm, the basis vector that minimizes the sum of squares of the residuals is selected. Due to orthogonality, this incremental selection is equivalent to evaluating the contributions of all possible subsets of vectors at the same time, ensuring that the greedy selection is consistent with the global optimum. Then, we have the following the key theorem of SOR, which plays the role of the second filtering strategy of basis vectors, and ensuring obtaining the optimal basis vectors.*

**Theorem 1.** *(Stepwise Orthogonal Regression) If the set of basis vectors $\{\theta_1^*, \theta_2^*, \ldots, \theta_n^*\}$ is orthogonal, then the greedy stepwise optimal selection of basis vectors will yield the same result as the global optimal solution.*

**Remark 2.** *SOR obtains the same result as the global optimum by sequentially selecting orthogonal basis vectors. It independently evaluate the contribution of each basis vectors to the model at each step of the greedy algorithm, selecting the basis vectors that minimizes the sum of squared residuals. The orthogonality ensures that this incremental selection is equivalent to simultaneously evaluating the contributions of all possible subsets of vectors, thus guaranteeing consistency with the global optimum. The insight is to use Stepwise Orthogonal Regression to filter out significant basis vectors from the initial set and eliminate irrelevant ones, ultimately obtaining the optimal basis vectors.*

The proof of Lemma 2, Lemma 3, and Theorem 1 are present in Appendix F.1, Appendix F.2, and Appendix F.3, separately. Thus, the stepwise optimal basis vectors selection is feasible.

## 4 EXPERIMENTS

To evaluate the performance of iQR, especially the demand satisfaction rate on resource scheduling tasks, we conduct extensive experiments on eight real-world time series benchmarks, and compare the performance with recent state-of-the-art (SOTA) methods.

**Datasets** Experiments are performed on various domains of datasets, covering electricity, energy, exchange and resource. All datasets are split into training, validation and test set chronologically with the ratio $7 : 1 : 2$. For fair comparison, the datasets are normalized with the z-score normalization. The details regarding datasets and experimental settings are provided in Appendix D.1.

**Baselines** A comprehensive comparison of effectiveness is performed against 7 baselines, including the SOTA long-term forecasting models TimeMixer (Wang et al., 2024) and PatchTST (Nie et al., 2023), the SOTA short-term forecasting models TimesNet (Wu et al., 2023) and FEDformer (Zhou et al., 2022), as well as other representative and competitive models, i.e., BasisFormer (Ni et al., 2023), iTransformer (Liu et al., 2023) and DLinear (Zeng et al., 2023). Please refer to Appendix D.2 for additional implementation details of baselines.

**Implementation Details** iQR is implemented in Python on a CPU environment, yet other experiments are performed with a single NVIDIA RTX A6000 GPU using PyTorch. All baselines are trained with a learning rate of $10^{-4}$ and the same quantile loss. We take Positive Mean Squared Errors ($PMSE = \sum_{i=1}^{N}(\hat{y} - y)^2 \cdot \mathbb{I}(\hat{y} \geq y)$), Positive Mean Absolute Errors ($PMAE = \sum_{i=1}^{N}|\hat{y} - y| \cdot \mathbb{I}(\hat{y} \geq y)$), and Quantile Regression Errors ($QRE = \frac{1}{N}\sum_{i=1}^{N}\mathbb{I}(\hat{y} \geq y)$) as the evaluation metrics. For more evaluation details, please refer to Appendix D.3 and Appendix D.4.

## 4.1 MAIN RESULTS

Resource scheduling is crucial in electricity, weather, and exchange rates as it optimizes resource allocation, and ensures system reliability. Resource datasets are usually multivariate time series and are considered as multichannel signals. PatchTST (Nie et al., 2023) has shown the effectiveness of channel-independent models. The experiments of iQR are conducted individually on each channel. For a better comparison, we follow the experimental setups in TimeMixer (Wang et al., 2024).

To assess the satisfaction rate in business demand, we conduct extensive resource scheduling experiments on eight datasets. Primary evaluation metric is the proportion of predictions exceeding the ground truth, also known as QRE. As shown in Table 1, iQR achieves the SOTA performance on all benchmarks, involving various time series with different frequencies and domains. Notably, iQR surpasses the latest model TimeMixer with an average QRE improvement of 3.53%. Specifically, iQR significantly outperforms the top-performing model Basisformer, with an 6.13% QRE increase in the weather dataset and a 3.52% QRE increase in the CPU dataset.

Additionally, scheduling CPU resources in advance is crucial for network services in parks. To meet the requirements of daily tasks, the allocated CPU resources must exceed the potential demand. As shown in Table 1, iQR achieves a higher satisfaction rate than TimeMixer on the CPU dataset, with an average increase of 3.98%. Note that iQR always meets the demands under different quantile setups. This further demonstrates the effectiveness of iQR, particularly in resource scheduling scenarios. The visualization results on three datasets are depicted in Figure 2. The local identification results shows the effectiveness of our scheduling strategy.

Table 1: QRE results for resource scheduling on eight datasets with historical length $H = 96$ and prediction length $P \in \{96, 192, 336, 720\}$. The historical and prediction length are set to $H = 36$ and $P \in \{24, 36, 48, 60\}$ for ILI. The ratio presents satisfaction rate of business demands. The best result are highlighted in **red bold**, and the second best results are marked with a blue underline.

| Methods | | τ | IMP. 0.9 | 0.95 | 1 | Ours 0.9 | 0.95 | 1 | TimeMixer 0.9 | 0.95 | 1 | TimesNet 0.9 | 0.95 | 1 | Basisformer 0.9 | 0.95 | 1 | iTransformer 0.9 | 0.95 | 1 | PatchTST 0.9 | 0.95 | 1 | Dlinear 0.9 | 0.95 | 1 | FEDformer 0.9 | 0.95 | 1 |
|---|---|---|---|---|---|---|---|---|---|---|---|---|---|---|---|---|---|---|---|---|---|---|---|---|---|---|---|---|---|
| ETTh1 | | 96 | 0.3% | 0.4% | 0.2% | **0.904** | **0.953** | **1.0** | 0.884 | 0.938 | 1.0 | 0.884 | 0.938 | 0.972 | 0.901 | 0.949 | 1.0 | 0.887 | 0.943 | 0.998 | 0.874 | 0.929 | 1.0 | 0.66 | 0.678 | 0.674 | 0.771 | 0.867 | 1.0 |
| | | 192 | 1.5% | 0.7% | 0.3% | **0.904** | **0.955** | **1.0** | 0.872 | 0.938 | 1.0 | 0.885 | 0.928 | 0.95 | 0.891 | 0.948 | 1.0 | 0.887 | 0.941 | 0.997 | 0.876 | 0.933 | 1.0 | 0.639 | 0.662 | 0.658 | 0.793 | 0.889 | 1.0 |
| | | 336 | 1.8% | 0.3% | 0.5% | **0.9** | **0.957** | **1.0** | 0.875 | 0.938 | 1.0 | 0.849 | 0.907 | 0.941 | 0.884 | 0.954 | 1.0 | 0.883 | 0.939 | 0.995 | 0.864 | 0.93 | 1.0 | 0.604 | 0.641 | 0.637 | 0.866 | 0.926 | 1.0 |
| | | 720 | 1.9% | −0.2% | 1.4% | **0.902** | 0.95 | **1.0** | 0.852 | 0.919 | 1.0 | 0.786 | 0.87 | 0.918 | 0.885 | 0.952 | 1.0 | 0.846 | 0.917 | 0.986 | 0.827 | 0.896 | 1.0 | 0.676 | 0.605 | 0.601 | 0.852 | 0.914 | 1.0 |
| ETTh2 | | 96 | 6.2% | 1.1% | 0.3% | **0.955** | **0.954** | **1.0** | 0.82 | 0.933 | 1.0 | 0.831 | 0.921 | 0.949 | 0.899 | 0.943 | 1.0 | 0.889 | 0.941 | 0.997 | 0.872 | 0.941 | 1.0 | 0.741 | 0.765 | 0.761 | 0.878 | 0.944 | 1.0 |
| | | 192 | 6.9% | 0.6% | 0.5% | **0.949** | **0.965** | **1.0** | 0.846 | 0.933 | 1.0 | 0.82 | 0.909 | 0.933 | 0.884 | 0.938 | 1.0 | 0.883 | 0.938 | 0.995 | 0.873 | 0.921 | 1.0 | 0.729 | 0.745 | 0.741 | 0.888 | 0.959 | 1.0 |
| | | 336 | 7.7% | 0.4% | 0.6% | **0.961** | **0.953** | **1.0** | 0.786 | 0.943 | 1.0 | 0.822 | 0.893 | 0.914 | 0.888 | 0.901 | 1.0 | 0.876 | 0.93 | 0.994 | 0.838 | 0.915 | 1.0 | 0.723 | 0.732 | 0.728 | 0.892 | 0.949 | 1.0 |
| | | 720 | 8.0% | 4.1% | 0.8% | **0.967** | **0.99** | 0.99 | 0.804 | 0.923 | 1.0 | 0.845 | 0.894 | 0.913 | 0.857 | 0.924 | 1.0 | 0.871 | 0.934 | 0.992 | 0.892 | 0.931 | 1.0 | 0.76 | 0.722 | 0.712 | 0.895 | 0.951 | 1.0 |
| ETTm1 | | 96 | 3.0% | 1.0% | 0.1% | **0.901** | **0.951** | **1.0** | 0.851 | 0.925 | 1.0 | 0.808 | 0.893 | 0.998 | 0.875 | 0.942 | 1.0 | 0.864 | 0.928 | 0.999 | 0.866 | 0.925 | 1.0 | 0.86 | 0.892 | 0.9 | 0.844 | 0.905 | 1.0 |
| | | 192 | 1.9% | 0.1% | 0.1% | **0.907** | **0.951** | **1.0** | 0.866 | 0.932 | 1.0 | 0.832 | 0.909 | 0.996 | 0.89 | 0.95 | 1.0 | 0.868 | 0.93 | 0.999 | 0.842 | 0.925 | 1.0 | 0.854 | 0.885 | 0.892 | 0.845 | 0.908 | 1.0 |
| | | 336 | 0.4% | 0.7% | 0.1% | **0.907** | **0.955** | **1.0** | 0.856 | 0.922 | 1.0 | 0.861 | 0.915 | 0.995 | 0.903 | 0.948 | 1.0 | 0.868 | 0.927 | 0.999 | 0.856 | 0.918 | 1.0 | 0.843 | 0.876 | 0.881 | 0.846 | 0.922 | 1.0 |
| | | 720 | 2.4% | 0.7% | 0.1% | **0.907** | **0.956** | **1.0** | 0.866 | 0.936 | 1.0 | 0.87 | 0.93 | 0.993 | 0.886 | 0.949 | 1.0 | 0.868 | 0.926 | 0.999 | 0.85 | 0.925 | 1.0 | 0.865 | 0.862 | 0.864 | 0.82 | 0.893 | 1.0 |
| ETTm2 | | 96 | 3.0% | 0.9% | 0.1% | **0.927** | **0.967** | **1.0** | 0.852 | 0.92 | 1.0 | 0.87 | 0.93 | 0.991 | 0.9 | 0.946 | 1.0 | 0.888 | 0.939 | 0.998 | 0.852 | 0.931 | 0.999 | 0.881 | 0.955 | 0.971 | 0.771 | 0.958 | 1.0 |
| | | 192 | 3.9% | 0.5% | 0.1% | **0.935** | **0.96** | **1.0** | 0.887 | 0.93 | 1.0 | 0.873 | 0.926 | 0.989 | 0.9 | 0.939 | 1.0 | 0.883 | 0.935 | 0.997 | 0.856 | 0.931 | 0.999 | 0.893 | 0.955 | 0.967 | 0.793 | 0.96 | 1.0 |
| | | 336 | 4.3% | −0.7% | 0.1% | **0.94** | 0.961 | **1.0** | 0.864 | 0.925 | 1.0 | 0.863 | 0.926 | 0.987 | 0.901 | 0.943 | 1.0 | 0.881 | 0.933 | 0.997 | 0.877 | 0.936 | 0.999 | 0.891 | 0.954 | 0.964 | 0.866 | 0.968 | 1.0 |
| | | 720 | 8.1% | 1.6% | 0.1% | **0.959** | **0.982** | **1.0** | 0.861 | 0.947 | 1.0 | 0.87 | 0.916 | 0.984 | 0.886 | 0.929 | 1.0 | 0.886 | 0.937 | 0.996 | 0.85 | 0.925 | 0.999 | 0.887 | 0.947 | 0.954 | 0.852 | 0.967 | 1.0 |
| Exchange | | 96 | −4.3% | −2.2% | 1.6% | 0.9 | 0.956 | **1.0** | 0.917 | 0.969 | 1.0 | 0.872 | 0.805 | 0.81 | 0.949 | 0.973 | 1.0 | 0.917 | 0.96 | 0.984 | 0.907 | 0.968 | 1.0 | 0.598 | 0.652 | 0.65 | 0.99 | 0.994 | 1.0 |
| | | 192 | −3.3% | −1.9% | 2.9% | 0.91 | 0.977 | **1.0** | 0.906 | 0.944 | 1.0 | 0.873 | 0.739 | 0.745 | 0.957 | 0.966 | 1.0 | 0.916 | 0.951 | 0.972 | 0.93 | 0.965 | 1.0 | 0.53 | 0.596 | 0.593 | 0.989 | 0.996 | 1.0 |
| | | 336 | −3.0% | −1.6% | 4.0% | 0.914 | 0.956 | **1.0** | 0.932 | 0.983 | 1.0 | 0.83 | 0.837 | 0.836 | 0.965 | 0.979 | 1.0 | 0.925 | 0.951 | 0.962 | 0.928 | 0.968 | 1.0 | 0.363 | 0.524 | 0.519 | 0.994 | 0.999 | 1.0 |
| | | 720 | −2.0% | −0.5% | 2.9% | 0.913 | 0.953 | **1.0** | 0.965 | 0.981 | 1.0 | 0.862 | 0.852 | 0.851 | 0.978 | 0.994 | 1.0 | 0.961 | 0.97 | 0.972 | 0.965 | 0.99 | 1.0 | 0.652 | 0.353 | 0.343 | 0.998 | 0.999 | 1.0 |
| Weather | | 96 | 0.1% | 3.4% | 0.2% | **0.905** | **0.966** | **1.0** | 0.83 | 0.889 | 1.0 | 0.817 | 0.857 | 0.955 | 0.832 | 0.912 | 1.0 | 0.838 | 0.905 | 0.991 | 0.864 | 0.913 | 0.998 | 0.904 | 0.934 | 0.936 | 0.894 | 0.856 | 1.0 |
| | | 192 | 2.6% | 2.8% | 0.2% | **0.92** | **0.951** | **1.0** | 0.837 | 0.907 | 1.0 | 0.725 | 0.835 | 0.944 | 0.836 | 0.898 | 1.0 | 0.835 | 0.899 | 0.99 | 0.855 | 0.901 | 0.998 | 0.897 | 0.925 | 0.926 | 0.859 | 0.845 | 1.0 |
| | | 336 | 0.6% | 4.5% | 0.2% | **0.904** | **0.958** | **1.0** | 0.825 | 0.882 | 1.0 | 0.708 | 0.839 | 0.934 | 0.83 | 0.897 | 1.0 | 0.835 | 0.887 | 0.989 | 0.825 | 0.907 | 0.998 | 0.899 | 0.917 | 0.917 | 0.854 | 0.906 | 1.0 |
| | | 720 | −1.3% | 1.9% | 0.3% | 0.909 | **0.952** | **1.0** | 0.833 | 0.899 | 1.0 | 0.737 | 0.812 | 0.917 | 0.839 | 0.898 | 1.0 | 0.831 | 0.888 | 0.987 | 0.844 | 0.905 | 0.997 | 0.921 | 0.906 | 0.905 | 0.827 | 0.934 | 1.0 |
| ILI | | 24 | 9.7% | 9.8% | 1.1% | **0.905** | **0.967** | **1.0** | 0.805 | 0.876 | 1.0 | 0.807 | 0.566 | 0.596 | 0.818 | 0.881 | 1.0 | 0.658 | 0.567 | 0.698 | 0.825 | 0.877 | 0.989 | 0.486 | 0.44 | 0.488 | 0.753 | 0.77 | 0.963 |
| | | 36 | 10.3% | 3.6% | 1.0% | **0.901** | **0.956** | **1.0** | 0.807 | 0.923 | 1.0 | 0.776 | 0.603 | 0.611 | 0.817 | 0.89 | 1.0 | 0.623 | 0.619 | 0.682 | 0.789 | 0.902 | 0.99 | 0.477 | 0.461 | 0.482 | 0.741 | 0.793 | 0.967 |
| | | 48 | 7.8% | 3.9% | 1.2% | **0.902** | **0.95** | **1.0** | 0.779 | 0.898 | 1.0 | 0.837 | 0.592 | 0.589 | 0.742 | 0.9 | 1.0 | 0.633 | 0.676 | 0.628 | 0.824 | 0.914 | 0.988 | 0.461 | 0.477 | 0.461 | 0.729 | 0.824 | 0.97 |
| | | 60 | 15.3% | 2.9% | 1.9% | **0.912** | **0.95** | **1.0** | 0.789 | 0.923 | 1.0 | 0.764 | 0.61 | 0.566 | 0.791 | 0.923 | 1.0 | 0.553 | 0.685 | 0.578 | 0.775 | 0.911 | 0.981 | 0.44 | 0.487 | 0.44 | 0.673 | 0.829 | 0.964 |
| CPU | | 96 | 2.4% | 1.7% | 0.3% | **0.906** | **0.952** | **1.0** | 0.849 | 0.929 | 1.0 | 0.877 | 0.911 | 0.979 | 0.885 | 0.936 | 1.0 | 0.824 | 0.889 | 0.997 | 0.864 | 0.914 | 1.0 | 0.711 | 0.741 | 0.738 | 0.841 | 0.846 | 1.0 |
| | | 192 | 5.5% | 3.4% | 0.2% | **0.905** | **0.953** | **1.0** | 0.848 | 0.916 | 1.0 | 0.858 | 0.922 | 0.971 | 0.857 | 0.915 | 1.0 | 0.82 | 0.856 | 0.998 | 0.857 | 0.91 | 1.0 | 0.67 | 0.714 | 0.712 | 0.842 | 0.876 | 1.0 |
| | | 336 | 5.1% | 1.9% | 0.2% | **0.907** | **0.955** | **1.0** | 0.812 | 0.926 | 1.0 | 0.863 | 0.937 | 0.97 | 0.836 | 0.895 | 1.0 | 0.779 | 0.804 | 0.998 | 0.81 | 0.888 | 1.0 | 0.566 | 0.672 | 0.671 | 0.825 | 0.839 | 1.0 |
| | | 720 | 7.9% | 7.1% | 0.1% | **0.912** | **0.981** | **1.0** | 0.833 | 0.916 | 1.0 | 0.845 | 0.886 | 0.929 | 0.845 | 0.907 | 1.0 | 0.742 | 0.762 | 0.999 | 0.813 | 0.886 | 1.0 | 0.736 | 0.569 | 0.566 | 0.778 | 0.834 | 1.0 |
| Ratio | | | - | | | 100% | 100% | 100% | 12.5% | 9.4% | 100% | 0% | 0% | 0% | 28.1% | 21.9% | 100% | 12.5% | 12.5% | 0% | 12.5% | 12.5% | 62.5% | 6.3% | 12.5% | 0% | 0.1% | 31.3% | 87.5% |
| Count | | | - | | | 89 | | | 36 | | | 6 | | | 65 | | | 20 | | | 34 | | | 9 | | | 48 | | |

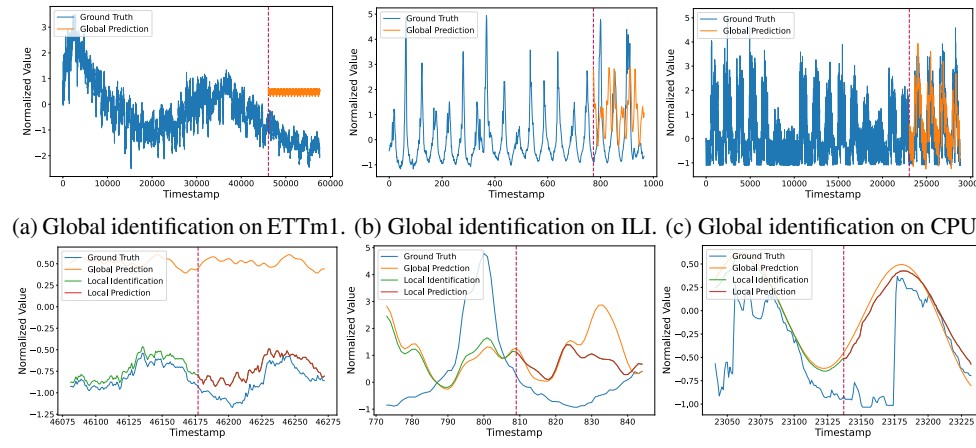

(a) Global identification on ETTm1. (b) Global identification on ILI. (c) Global identification on CPU.

(d) Local identification on ETTm1. (e) Local identification on ILI. (f) Local identification on CPU.

Figure 2: Visualization of identifications on three datasets. The historical and future sequence length are set to 96 for ETTm1 and CPU dataset, while 36 for ILI dataset. The quantile $\tau$ is 0.9, indicating that over $90\%$ of the identifications are expected to exceed the ground truth.

Furthermore, we have conducted an evaluation of the performance across various quantiles, and the comprehensive results of this analysis can be accessed in Appendix D.5. The findings indicate that iQR is adept at fulfilling business demands while incurring significantly lower costs.

## 4.2 MODEL ANALYSIS

**Ablation Studies**  To verify the design of each component in the scheduling framework iQR, we conducted a comprehensive ablation study. The purpose is systematically isolating and evaluating the contribution of the basis vectors selection SOR, the STL, the knowledge fusion technology, and the lasso penalized quantile regression.

Table 2: Ablation studies on the ETTh1 dataset with $H = 96$, $P = 96$, and $\tau = 0.9$. The best result are highlighted in **red bold**, and the second best results are marked with a blue underline.

| Tasks | iQR | iQR-R | iQR-S | iQR-K | iQR-L | iQR-Q |
|-------|-----|-------|-------|-------|-------|-------|
| PMSE | **1.185** | 1.191 | 1.19 | 0.909 | 0.561 | 0.558 |
| PMAE | **0.782** | 0.787 | 0.785 | 0.697 | 0.574 | 0.465 |
| QRE | **0.904** | 0.901 | 0.9 | 0.845 | 0.736 | 0.455 |

Five modified versions are presented, each lacking a key element: (1) model without SOR, denoted as iQR-R, (2) model excludes the STL, marked as iQR-S, (3) model omits the knowledge fusion technology, represented as iQR-K, (4) model without the lasso penalty, termed as iQR-L, and (5) model replaces quantile regression with linear regression, labeled as iQR-Q. The experiments are conducted on ETTh1 dataset.

Table 2 illustrates the performance among variants. iQR outperforms other baselines on all three evaluation metrics, suggesting that it is effective for resource scheduling tasks.

**iQR vs. Maximum Value Baseline**  We employ a simple and intuitive evaluation method, the maximum value baseline. Reference line are plotted based on the maximum value observed in the training dataset. The lines symbolize an ideal state where future ground truth is always lower than the maximum values. As shown in Table 3, iQR achieves significant improvement in performance, with an average reduction of $80.49\%$ in PMSE and $65.55\%$ in PMAE.

## 4.3 EFFICIENCY ANALYSIS

**Complexity Analysis**  The spatial complexity is $\mathcal{O}(t \cdot d_y)$, and the temporal complexity is $\mathcal{O}((2n + 3n \cdot n_{batch}) \cdot d_y)$, where $t$ is the timestamps, $d_y$ is the number of features, and $n_{batch}$ is the number of batches in testing phase. Due to channel independence, the model identifies and predicts each feature separately. Specifically, if the dataset contains only one feature, the time complexity of constructing the global basis vectors during the training phase is $\mathcal{O}(2K^2)$, which is omitted since $K \ll t$; the

Table 3: Resource scheduling results for iQR and Maximum Value Baseline on ETTh1 datasets with historical length $H = 96$ and prediction length $P \in \{96, 192, 336, 720\}$. The improvement is noted as IMP. and is displayed in the end of the table.

| Length | | 96 | | | 192 | | | 336 | | | 720 | |
|---|---|---|---|---|---|---|---|---|---|---|---|---|
| Metrics | PMSE | PMAE | QRE | PMSE | PMAE | QRE | PMSE | PMAE | QRE | PMSE | PMAE | QRE |
| ours | 2.348 | 1.282 | 1.0 | 2.311 | 1.279 | 1.0 | 2.46 | 1.309 | 1.0 | 2.579 | 1.348 | 1.0 |
| Maximum Value | 12.412 | 3.316 | 1.0 | 12.198 | 3.275 | 1.0 | 12.436 | 3.304 | 1.0 | 12.654 | 3.343 | 1.0 |
| IMP. | 81.08% | 61.34% | 0% | 81.05% | 60.95% | 0% | 80.22% | 60.38% | 0% | 79.62% | 59.68% | 0% |

time complexity of the $L_1$QR and SOR algorithms are $\mathcal{O}(n)$. During the testing phase, both the STL algorithm, the SOR algorithm, and the $L_1$QR algorithm have a time complexity of $O(n)$, and the time complexity of constructing local basis vectors can be omitted. Thus, the total time complexity during testing is $O(3n)$. Additionally, the model has fewer parameters due to the absence of neural networks, and the spatial complexity can be approximated as the data scale $t \cdot d_y$.

**Memory Cost and Running Time**    We compare the GPU memory cost and running time against the current SOTA baselines, covering MLP-based TimeMixer (Wang et al., 2024) and Transformer-based iTransformer (Liu et al., 2023), etc. In Figure 3, iQR outperforms other baselines in terms of running time and memory usage. Specifically, iQR is much faster for various historical sequence length $H$ from 96 to 2880, with the prediction sequence length is fixed at 96. Moreover, Figure 3(b) show that iQR is lightweight. This further demonstrates the advantages of iQR in terms of resource utilization and time efficiency. Additional efficiency analysis results are presented in Appendix D.8.

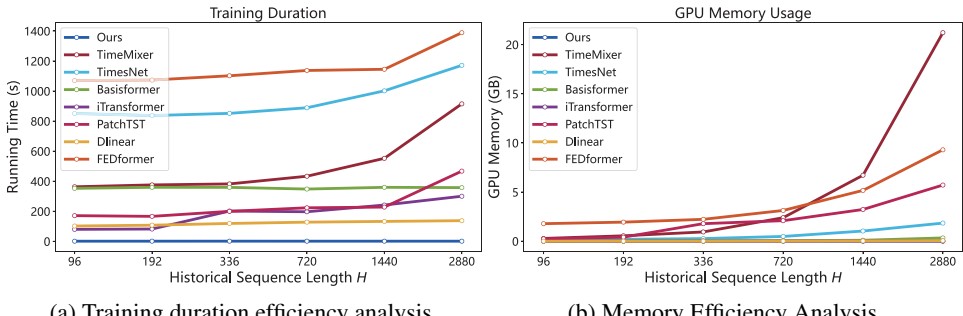

(a) Training duration efficiency analysis.    (b) Memory Efficiency Analysis.

Figure 3: Efficiency analysis of eight algorithms. The results are derived from the experiments on the ETTh1 dataset, and we report the average result from ten experiments.

## 5 CONCLUSION REMARKS

In this study, we address the crucial issue of resource scheduling across diverse industrial contexts by introducing a novel framework, termed iQR, which leverages sparse system identification techniques. This framework deviates significantly from conventional model-based or prediction-based scheduling approaches, adopting instead a global-local coordination strategy to bolster identification performance. Specifically, we first utilize Fast Fourier Transform (FFT) to comprehensively extract vital features from a frequency domain perspective. Furthermore, iQR incorporates Stepwise Orthogonal Regression (SOR) to derive more prominent basis vectors and employs $L_1$-norm Quantile Regression ($L_1$QR) to ensure precise scheduling across various quantile levels. The deployment of $L_1$QR for resource scheduling not only fulfills resource demands with high confidence but also minimizes resource allocation costs. Rigorous experiments conducted on eight datasets demonstrate that iQR attains state-of-the-art (SOTA) performance, characterized by a higher satisfaction rate and reduced economic cost, compared to recent deep learning-based prediction methods incorporating quantile regression. Notably, iQR is lightweight, with training durations measured in seconds, and operates efficiently without the need for GPU resources. Looking ahead, we aim to broaden the applicability of iQR to a wider range of scheduling scenarios and explore more potent and efficient solutions for time series analysis in resource-constrained environments.

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

# A NOTATIONS

A comprehensive table of all notations used in this work is provided here. The following tables serve as a reference to clarify the meaning and usage of each symbol throughout this study.

**Numbers and Arrays**

| | |
|---|---|
| $x$ | A scalar |
| $X$ | A vector |
| $\mathbf{X}$ | A matrix |
| $\text{diag}(\boldsymbol{a})$ | A square, diagonal matrix with diagonal entries given by $\boldsymbol{a}$ |
| $\beta$ | The vector of regression coefficients to be estimated |
| $\lambda$ | The regularization parameter. A sparse solution is obtained when $\lambda$ is large; otherwise, more features are retained |
| $\tau$ | The quantile $\tau$ denotes the ratio that $100 \times \tau\%$ of predictions are greater than the ground truth, $\tau \in [0, 1]$ |
| $\mathbf{w}$ | The weighted vector, $w_i = (\alpha^{1/\gamma} + b\Delta)^\gamma$, for $b = 0, \ldots, L_{batch} - 1$ and $\Delta = (\xi^{1/\gamma} - \alpha^{1/\gamma})/(L_{batch} - 1)$ |
| $\mathbf{T}$ | The significant periods for global identification, $\mathbf{T} = [T_1, T_2, \ldots, T_n]$ |

**Sets**

| | |
|---|---|
| $\mathbb{R}$ | The set of real numbers |
| $\boldsymbol{\Theta}$ | The set of basis vectors |
| $\{0, 1, 2, 3\}$ | The set containing 0, 1, 2, and 3 |

**Indexing**

| | |
|---|---|
| $\mathcal{B}_b$ | The $b$th test batch, the test dataset is split into batches |
| $X^i$ | The values of $\mathbf{X}$ at feature dimension $i$ |
| $\hat{X}^i$ | The input sequence after elimination of outliers by knowledge fusion strategy, $\hat{X}^i = \mathbf{w}X^i + (1 - \mathbf{w})\hat{Y}g^i$ |
| $\tilde{X}^i_{l_b,t}$ | The output sequence for the $b$th test batch $\mathcal{B}b$ at feature dimension $i$ derived by STL |
| $\mathbf{X}_t$ | The historical sequence with a look back window of $H$, $\mathbf{X}_t = [X_{t-H+1}, X_{t-H+2}, \ldots, X_t] \in \mathbb{R}^{H \times d_x}$ |
| $Y^i$ | The values of $\mathbf{Y}$ at feature dimension $i$ |
| $Y_t$ | The multivariate values of $d_y$ distinct series at timestamp $t$, $Y_t \in \mathbb{R}^{d_y}$ |
| $Y^i_g$ | The global identification at feature dimension $i$ |
| $Y^i_{l_b,t}$ | The local identification for the $b$th test batch $\mathcal{B}_b$ at feature dimension $i$ |
| $\mathbf{Y}_t$ | The future sequence with a horizon window of $P$, $\mathbf{Y}_t = [Y_{t+1}, Y_{t+2}, \ldots, Y_{t+P}] \in \mathbb{R}^{P \times d_y}$ |

| | |
|---|---|
| $\hat{\mathbf{Y}}_t$ | The prediction sequence with a horizon window of $P$, $\hat{\mathbf{Y}}_t = [\hat{Y}_{t+1}, \hat{Y}_{t+2}, \dots, \hat{Y}_{t+P}] \in \mathbb{R}^{P \times d_y}$ |
| $\mathbf{T}^i_{l_b,t}$ | The significant periods at feature dimension $i$ for a test batch |
| $\mathbf{\Theta}^i_g$ | The global basis vectors $\mathbf{\Theta}^i_g = [\sin(\frac{2\pi \mathbf{t}^i}{\mathbf{T}^i}), \cos(\frac{2\pi \mathbf{t}^i}{\mathbf{T}^i})]$ |
| $\hat{\mathbf{\Theta}}^i_g$ | The significant global basis vectors at feature dimension $i$ obtained by SOR filtering method |
| $\mathbf{\Theta}^i_{l_b,t}$ | The local basis vectors at feature dimension $i$ for the $b$th test batch $\mathcal{B}_b$, $\mathbf{\Theta}^i_{l_b,t} = [\sin(\frac{2\pi \mathbf{t}^i_{l_b,t}}{\mathbf{T}^i_{l_b,t}}), \cos(\frac{2\pi \mathbf{t}^i_{l_b,t}}{\mathbf{T}^i_{l_b,t}})]$ |
| $\hat{\mathbf{\Theta}}^i_{l_b,t}$ | The significant local basis vectors for the $b$th test batch $\mathcal{B}_b$ at feature dimension $i$ obtained by SOR filtering method |
| $\beta^i_g$ | The global regression coefficients vector at feature dimension $i$ to be estimated |
| $\beta^i_{l_b,t}$ | The local regression coefficients vector at feature dimension $i$ to be estimated |

**Functions**

| | |
|---|---|
| $||\boldsymbol{x}||_1$ | $L_1$ norm of $\boldsymbol{x}$ |
| $\mathbb{I}(\text{condition})$ | is 1 if the condition is true, 0 otherwise |
| $\rho_\tau(u)$ | The Koenker and Bassett check function, $\rho_\tau(u) = u(\tau - \mathbb{I}(u < 0))$ |

## B  PROBLEM VISUALIZATION AND PERFORMANCE ANALYSIS

To visually illustrate the problem, we set the quantile $\tau = 1$ and explain the following three figures. The grey-shaded areas present the difference between the prediction and the ground truth, also known as economic cost.

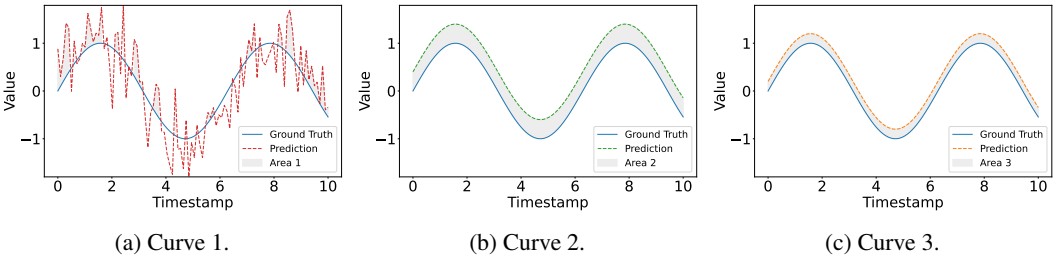

(a) Curve 1.  (b) Curve 2.  (c) Curve 3.

Figure 4: Comparison of three predictive model performance.

Figure 4(a) shows that 50% of the predicted values exceed the ground truth. This model is difficult to meet practical demands as it cannot guarantee that all predicted values will be greater than the actual demand, potentially leading to insufficient resources.

Figure 4(b) shows that 100% of the predicted values exceed the ground truth, which meets the requirements for resource scheduling. It effectively prevents task failure by ensuring adequate resource allocation. Therefore, the results in Figure 4(b) are better than in Figure 4(a).

Figure 4(c) shows another prediction curve. The grey area is smaller, indicating that the additional cost brought by the model, after meeting the requirements, is lower, meaning higher resource utilization efficiency. Thus, the results in Figure 4(c) are better than in Figure 4(a) and Figure 4(b).

## C   INTRODUCTION TO THE KEY COMPONENTS

**Sparse System Identification**   Sparse System Identification (SSI) (Brunton et al., 2016; Wilms et al., 2023; Fasel et al., 2022) infers the underlying dynamics and the algebraic representation of a control system from observed data. This methodology is rooted in the principle of parsimony, seeking to disentangle the most concise and informative representation of system behavior by selecting a minimal set of relevant basis vectors from a predefined library.

SSI is inherently characterized by the basis function library, which commonly encompasses polynomial series, trigonometric functions, and other suitable functional forms. Given this library, we are able to identify a sparse combination of basis vectors that best approximates the system dynamics. This is achieved by minimizing a sparsity-promoting penalty term within the optimization framework. SSI eliminates superfluous terms, thereby yielding compact and interpretable dynamical models.

SSI has demonstrated remarkable progress in revealing hidden system dynamics within data, resulting in highly accurate predictions in various contexts (Liu et al., 2024). However, the efficacy of SSI for business-driven identification prediction is not without challenges. Direct application of the method without tailored strategies for basis vectors design may lead to accumulation of errors over extended prediction horizons, compromising its suitability for long-range predictions.

**$L_1$-norm Quantile Regression**   Traditional regression analysis focuses on the relationship between covariates and the conditional expectation of the response variable. Models derived from such analyses, such as linear or generalized linear regressions, aim to construct a function that predicts the conditional mean of the response given observed covariate values. While these methods effectively capture the central tendency of the response distribution in many instances, they provide a single-dimensional view and can be susceptible to outliers or data heterogeneity. To address these limitations, Quantile Regression (QR) has emerged as an alternative approach, particularly when interest lies in other aspects of the response distribution.

Quantile regression extends the scope of conventional regression by investigating the association between covariates and conditional quantiles of the response variable rather than solely its conditional expectation. Specifically, quantile regression models estimate the conditional quantile of the dependent variable given the value of the independent variable. QR offers a comprehensive portrayal of the response distribution across different quantiles, encompassing tail behavior, asymmetry, and potential non-linear effects.

Recently, $L_1$-norm Quantile Regression ($L_1$QR) has been proposed as a robust and flexible approach for modeling conditional quantiles (Li & Zhu, 2008; Shang & Kong, 2021). $L_1$QR employs a penalty term based on the absolute values of the regression coefficients, thereby promoting sparsity in the model. Unlike the $L_2$-norm, some regression coefficients derived by the $L_1$-norm likely to be estimated as zero. This facilitates variable selection, particularly in high-dimensional settings, by identifying a subset of key covariates with significant influence on specific quantiles. For a given quantile $\tau$, the objective function of $L_1$QR is formulated as:

$$\min_{\beta} \sum_{i=1}^{n} \rho_\tau(Y^i - X^i\beta) + \lambda||\beta||_1,$$

where $y^i$ is the is the response value for the $i$th observation, $X^i$ represents the corresponding covariate values, $\beta$ is the vector of regression coefficients to be estimated, and $\rho_\tau(u)$ is the Koenker and Bassett check function (Koenker & Bassett Jr, 1978), that is

$$\rho_\tau(u) = \begin{cases} \tau \cdot u & \text{if } u > 0, \\ -(1-\tau) \cdot u & \text{otherwise.} \end{cases}$$

**Fourier Transformation**   The Fourier transform (FT) is a mathematical tool used to convert signals from the time domain to the frequency domain, revealing the underlying periodic components in a signal. Mathematically, the Fourier transform of $f(t)$ is given by:

$$F(\omega) = \int_{-\infty}^{\infty} f(t)e^{-j\omega t}dt,$$

where $\omega$ is the angular frequency, $j$ is the imaginary unit, and $e$ is the base of the natural logarithm. The transformed function $F(\omega)$ represents the amplitude and phase of each frequency component in the original signal, enabling the identification of dominant frequencies and their relative contributions to the overall signal structure. For digital signals or sampled time series data, the continuous-time Fourier transform is replaced by the Discrete Fourier Transform (DFT). The DFT computes the frequency-domain representation of a finite-length discrete-time signal $x_n$:

$$X_k = \sum_{n=0}^{N-1} x_n e^{-j2\pi kn/N}, \quad k = 0, 1, \ldots, N-1,$$

where $X_k$ represents the amplitude and phase of the $k$th frequency component in the discrete signal, and the angular frequency $\omega = 2\pi k/N$ corresponds to the $k$th harmonic of the fundamental frequency $\omega_0 = 2\pi/N$.

While the DFT is a powerful tool for frequency analysis, its computational complexity is $\mathcal{O}(n^2)$, which can become prohibitively expensive for large datasets. The Fast Fourier Transform (FFT) is an algorithmic innovation that significantly reduces the computational cost of computing the DFT, achieving a time complexity of $\mathcal{O}(n \log n)$. All waveforms are actually just the sum of simple sinusoids of different frequencies. The FFT exploits the symmetries and periodicity inherent in the DFT formula to compute the transform more efficiently, making it the preferred choice for practical applications involving large time series datasets.

**Stepwise Orthogonal Regression**   Stepwise Orthogonal Regression (SOR) is a sequential feature selection algorithm specifically tailored for model building in the context of time series analysis and forecasting. SOR combines the principles of forward selection and orthogonal least squares to efficiently identify and incorporate the most influential variables into a model.

The SOR algorithm proceeds in a stepwise manner, iteratively adding one predictor at a time to an initially empty model. At each iteration, the algorithm selects the variable that maximally improves the model fit while minimizing collinearity issues through the use of orthogonal projections. The improvement in model fit is typically assessed via a goodness-of-fit criterion, such as the reduction in the sum of squared errors or the increase in the coefficient of determination. Once a variable is added, its contribution is made orthogonal to the existing model terms, ensuring that subsequent selections are independent and do not merely replicate previously captured information. This process continues until a stopping criterion is met, which could be based on a predefined maximum number of predictors, a threshold for the improvement in model fit, or a desired level of explained variance.

In the realm of quantile regression for time series forecasting, SOR can be employed to select the most relevant predictors for each quantile level of interest. Given the potential heterogeneity in the effects of different variables across different quantiles, SOR enables the construction of separate, tailored models for each quantile, ensuring that each model includes only the most influential predictors for that specific quantile. As a result, SOR-enhanced quantile regression models provide more accurate and reliable forecasts across the entire conditional distribution of the time series.

**Seasonal-Trend decomposition using LOESS**   Seasonal-Trend decomposition using LOESS (STL) is a nonparametric filtering procedure, proposed by Cleveland (Cleveland et al., 1990), with LOcally Estimated Scatterplot Smoothing (LOESS) used for estimating nonlinear relationships. STL has gained popularity due to its ability to handle diverse types of time series data, including those exhibiting irregular seasonal patterns and time-varying trends. The simplicity of STL allows fast computation for long sequence. STL decomposes time series $X$ into three additive components, namely trend $T$, seasonal $S$, and residual $R$.

$$X = \mathcal{T} + \mathcal{S} + \mathcal{R},$$

in which the trend component $T$ captures the long-term behavior of a time series, embodying its overall direction and pattern over time. Seasonality $S$ refers to recurring patterns or cycles occurring within fixed time frames, such as daily, weekly, or annual patterns. The residual component $R$, also known as the remainder or irregular component, represents unexplained time series variations after removing the trend and seasonal components. It encapsulates random fluctuations, noise, or irregularities not accounted for by the trend and seasonal patterns.

STL decomposition serves as a critical preprocessing step in time series forecasting models. By decomposing the raw time series into its constituent parts, we gain deeper insights into the underlying structure and identify potential patterns and anomalies. This decomposition facilitates the development of more accurate prediction models by modeling each component separately and incorporate relevant features into the forecasting framework.

## D EXPERIMENTAL DETAILS

In this section, we provide a comprehensive introduction of experiments, including 8 real-world datasets, 7 baselines and 3 evaluation metrics.

### D.1 DATASETS

To evaluate the performance of prediction models, the experiments are conducted on 8 well-established datasets, involving Electricity, Exchange rate, Weather, Illness and CPU usage. The detailed description is presented in Table 4.

**Electricity Transformer Temperature (ETT) (Zhou et al., 2021)** It contains data from 2 power transformers in two regions of a province in China, recorded every minute (ETTm). Hourly data (ETTh) is also available. The data includes 8 characteristics, including date, target predicted value "oil temperature" and 6 different types of external power load characteristics [1].

**Exchange (Lai et al., 2018)** It contains exchange rate data from 8 countries (Australia, British, Canada, China, Japan, New Zealand, Singapore, and Switzerland), recorded every day ranging from 1990 to 2016 [2].

**Weather** From a weather station of the Max Planck Biogeochemistry Institute in German, it aggregates 21 meteorological variables including humidity and temperature, sampled every 10 minutes throughout 2020 [3].

**ILI** It illustrates the ratio of patients presenting with influenza-like illness relative to the total patient population. The dataset encompasses weekly data sourced from the Centers for Disease Control and Prevention (CDC) of the United States, spanning from 2002 through 2021 [4].

**CPU** It collects the CPU usage from a park of Huawei Technologies Co., Ltd. The sampling interval is 1 minute, and the dataset spans from May 8th to May 27th, 2023.

Table 4: Summary of real-world datasets.

| Dataset | Frequency | Dim. | Series Length | Dataset Size | Information |
|---------|-----------|------|---------------|--------------|-------------|
| ETTh1 | Hourly | 7 | $\{96, 192, 336, 720\}$ | $(10080, 1440, 2880)$ | Electricity |
| ETTh2 | Hourly | 7 | $\{96, 192, 336, 720\}$ | $(10080, 1440, 2880)$ | Electricity |
| ETTm1 | 15min | 7 | $\{96, 192, 336, 720\}$ | $(40320, 5760, 11520)$ | Electricity |
| ETTm1 | 15min | 7 | $\{96, 192, 336, 720\}$ | $(40320, 5760, 11520)$ | Electricity |
| Exchange | Daily | 8 | $\{96, 192, 336, 720\}$ | $(5120, 665, 1422)$ | Exchange rate |
| Weather | 10min | 21 | $\{96, 192, 336, 720\}$ | $(36792, 5271, 10540)$ | Weather |
| ILI | Weekly | 7 | $\{24, 36, 48, 60\}$ | $(617, 74, 170)$ | National illness |
| CPU | 1min | 1 | $\{96, 192, 336, 720\}$ | $(20160, 2880, 5760)$ | CPU usage |

---

[1] https://github.com/zhouhaoyi/ETDataset
[2] https://github.com/laiguokun/multivariate-time-series-data
[3] https://www.bgc-jena.mpg.de/wetter/
[4] https://gis.cdc.gov/grasp/fluview/fluportaldashboard.html

## D.2 BASELINES

Time series forecasting modeling is a long-standing research problem. From early statistical models to later deep models, many new models have been proposed for sequence modeling and time series forecasting. However, long-term forecasting still suffers from a lack of accuracy. Considering the challenges of traditional statistical methods in long-term forecasting, the baselines we compare are mainly neural network methods. Specifically, we consider the latest time series forecasting methods, including TimeMixer (Wang et al., 2024), TimesNet (Wu et al., 2023), Basisformer (Ni et al., 2023), iTransformer (Liu et al., 2023), PatchTST (Nie et al., 2023), Dlinear (Zeng et al., 2023), and FEDformer (Zhou et al., 2022). The details are introduced as follows

**TimeMixer**  TimeMixer is an architecture based entirely on multilayer perceptual machines. It consists of past decoupled mixing and future multi-predictor mixing modules, which decompose multi-scale sequences and integrate multiple predictors for prediction, respectively.TimeMixer provides insights, that is, time series exhibit different patterns at different sampling scales. Its scheduling implementation is public available at `https://github.com/kwuking/TimeMixer`.

**TimesNet**  TimesNet is an innovative time series analysis method that converts a one-dimensional series into a multi-periodic two-dimensional tensor, which maps the time series into a two-dimensional space and enhances the representation of periodic features. It decomposes complex temporal dynamics into intra-periodic fluctuations and inter-periodic trends, and thus performs better on datasets with significant periodic structure. Code is available at the repository `https://github.com/thuml/Time-Series-Library`.

**Basisformer**  BasisFormer is an end-to-end time series forecasting architecture that designs learnable and interpretable substrates for forecasting. It consists of three core components: acquiring the substrate through adaptive self-supervised learning; computing the similarity coefficients between the time series and the substrate in historical perspective through a two-way cross-attention mechanism; and selecting the substrate based on the similarity coefficients to generate accurate future forecasts. The source code can be found at `https://github.com/nzl5116190/Basisformer`.

**iTransformer**  iTransformer is an advanced model designed for time series forecasting. Utilizing an "inverted Transformer" architecture, iTransformer has demonstrated great effectiveness in forecasting time series data, especially for long term forecasting problems with complex periodicity, trends, and non-stationarity. The source code is available at the repository `https://github.com/thuml/iTransformer`.

**PatchTST**  PatchTST is a powerful Transformer-based model specialized in multivariate time series prediction and self-supervised representation learning. There are two core components: (i) segmentation of the time series into subsequence patches, which are then provided to the Transformer; and (ii) channel independence, where each channel represents a univariate time series with shared embeddings and Transformer weights for all series.The components in PatchTST greatly improve the accuracy of long-term forecasts. Code of PatchTST is present at the repositry `https://github.com/yuqinie98/PatchTST`.

**Dlinear**  DLinear represents a straightforward neural network prediction model that ingeniously integrates time series decomposition with linear layers, unlike Tranformer-based models. The essence of DLinear lies in its direct multi-step forecasting approach, decomposing historical time series data into trend and remainder components, each modeled through separate single-layer linear networks to achieve predictive outputs. The source code for DLinear is openly accessible at `https://github.com/cure-lab/LTSF-Linear`.

**FEDformer**  FEDformer is a Transformer-based model that is designed to tackle the long-term time series prediction issue by incorporating Fourier transforms and wavelet basis vectors to enhance the effectiveness of the model. The time complexity of FEDformer is $\mathcal{O}(L)$, where $L$ is the length of the sequence, which means that FEDformer can significantly improve the computational efficiency when dealing with long sequences compared to traditional methods. Its implementation is provided in `https://github.com/MAZiqing/FEDformer`.

### D.3 EVALUATION METRICS

Due to the practical need of specific task, the business-driven identification prediction, we adopt three metrics in the experiments, including PMSE (Positive Mean Square Error), PMAE (Positive Mean Absolute Error), and QRE (Qutantile regression Error).

Concretely, given the ground truth of future sequence $\mathbf{Y}_t^i = [Y_{t+1}^i, Y_{t+2}^i, \ldots, Y_{t+P}^i]$ with $Y_t^i = [Y_{t+1}^i, Y_{t+2}^i, \ldots, Y_{t+P}^i]$, and the prediction sequence $\hat{\mathbf{Y}}_t^i = [\hat{Y}_{t+1}^i, \hat{Y}_{t+2}^i, \ldots, \hat{Y}_{t+P}^i]$ with $\hat{Y}_t^i = [\hat{Y}_{t+1}^i, \hat{Y}_{t+2}^i, \ldots, \hat{Y}_{t+P}^i]$, the metrics are given as follows

$$PMSE = \sum_{i=1}^{d_y} \sum_{j=t+1}^{t+P} (\hat{Y}_j^i - Y_j^i)^2 \cdot \mathbb{I}(\hat{Y}_j^i \geq Y_j^i),$$

$$PMAE = \sum_{i=1}^{d_y} \sum_{j=t+1}^{t+P} |\hat{Y}_j^i - Y_j^i| \cdot \mathbb{I}(\hat{Y}_j^i \geq Y_j^i), \tag{8}$$

$$QRE = \frac{1}{d_y \times P} \sum_{i=1}^{d_y} \sum_{j=t+1}^{t+P} \mathbb{I}(\hat{Y}_j^i \geq Y_j^i).$$

### D.4 IMPLEMENTATION DETAILS

All baselines models are trained with the QRE loss, using Adam optimizer with an initial learning rate of $10^{-4}$. Batchsize is set to 32 as default. If no reduction in validation loss is observed, training is stopped prematurely after 3 epochs. Then, the model with least validation loss is saved for further evaluation.

For fair comparison, we assign the same search space to the common parameters in each model. And the best performance under the search space is reported. The seed equals to 2024. The data segmentation ratio is 7:1:2, and since the validation set data is often used in the model training stage and there is no validation stage in our algorithm, it is 8:2 for our identification algorithm, shown in Figure 5. In this study, the adaptive sparse group lasso (asgl) Python package developed by Mendez-Civieta et al. (2021), is adopted for quantile regression analysis.

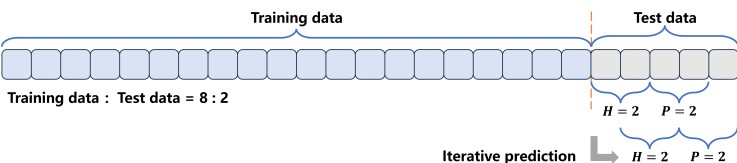

Figure 5: Data division and iterative identification schematic diagram.

Concretely, we adopted a uniform strategy in our experimental design, where the 6 methods (TimeMixer, TimesNet, iTransformer, PatchTST, DLinear, and FEDformer) employees the tools provided by the Python library `Time Series Library (TSlib)`[5] to perform the time series forecasting task. Moreover, the Basisformer model is performed with its official implementation.

### D.5 DETAILED RESULTS

For the resource scheduling task, we design an innovative approach, also known as iQR, that focuses on the Quantile Regression Error (QRE) as a pivotal performance metric. In order to investigate the model performance under different parameter settings, the loss function is set with different quantiles ($\tau = 1.0, 0.95, 0.9$). The primary goal is to evaluate whether the identified values consistently exceed the ground truth. The Positive Mean Square Error (PMSE) and Positive Mean Absolute Error (PMAE) of these predictions are then minimized under this constraint.

---

[5] https://github.com/thuml/Time-Series-Library

Under the tightest constraint, i.e., all predictions must exceed the actual values, our model iQR performs well. In several datasets such as ETT, Exchange, and Weather, there is still a low PMSE and PMAE for a QRE of 1. This suggests that not only are the predictions generally higher than the true values, but the error in these over-valued predictions (measured in terms of the PMSE and PMAE) is also minimized. For example, as shown in Table 5, in a dataset with a history length of 96 and predicted lengths varying from 96 to 720, the iQR model demonstrates high accuracy and low error, outperforming several competing methods including TimeMixer, and FEDformer.

Then, the constraint is relaxed to allow for a small percentage of predictions to be lower than the ground truth, the quantile is set to 0.95. iQR still maintains the high satisfaction rate regarding the business demand, with PMSE and PMAE lower than many competitive baselines. The results in Table 6 indicates that the model provides highly accurate predictions even existing a certain margin of error. The QRE is consistently higher than 0.95, demonstrating the robustness of designed algorithm.

Moreover, we further relax the constraints, that is, the quantile decreases to 0.9. iQR still satisfies the business demand, although exhibits the second best results on ETTh2 dataset. iQR achieves an optimal or suboptimal performance ratio of $92/96$ across evaluated metrics, illustrating its broad effectiveness. Notably, even the prediction constraints are relaxed, the model maintains a low predictive error, thereby highlighting its particular superiority in the resource scheduling tasks.

Table 5: Resource scheduling results on eight datasets with historical length $H = 96$ and prediction length $P \in \{96, 192, 336, 720\}$. The historical and prediction length are set to $H = 36$ and $P \in \{24, 36, 48, 60\}$ for ILI. Lower PMSE and PMAE for similar QRE indicate more accurate predictions and lower resource usage. If PMSE or PMAE is greater than 100, the symbol - is used to replace the original value. The quantile is set to 1. If the QRE in the result is less than the quantile, it will not be used for comparison. The ratio presents the percentage by which prediction exceeds ground truth. The best result are highlighted in **red bold**, and the second best results are marked with a blue underline.

| Methods | IMP. PMSE | IMP. PMAE | IMP. QRE | Ours PMSE | Ours PMAE | Ours QRE | TimeMixer PMSE | TimeMixer PMAE | TimeMixer QRE | TimesNet PMSE | TimesNet PMAE | TimesNet QRE | Basisformer PMSE | Basisformer PMAE | Basisformer QRE | iTransformer PMSE | iTransformer PMAE | iTransformer QRE | PatchTST PMSE | PatchTST PMAE | PatchTST QRE | Dlinear PMSE | Dlinear PMAE | Dlinear QRE | FEDformer PMSE | FEDformer PMAE | FEDformer QRE |
|---|---|---|---|---|---|---|---|---|---|---|---|---|---|---|---|---|---|---|---|---|---|---|---|---|---|---|---|
| ETTh1 96 | 78.6% | 51.4% | 0.0% | **3.26** | **1.61** | 1.0 | - | - | 1.0 | 3.66 | 1.53 | 0.97 | - | - | 1.0 | 16.01 | 3.31 | 1.0 | 79.24 | 7.5 | 1.0 | 0.49 | 0.49 | 0.67 | 15.26 | 3.73 | 1.0 |
| ETTh1 192 | 81.3% | 58.3% | 0.0% | **3.23** | **1.48** | 1.0 | - | - | 1.0 | 4.34 | 1.65 | 0.95 | - | - | 1.0 | 18.54 | 3.55 | 1.0 | 94.5 | 8.18 | 1.0 | 0.54 | 0.52 | 0.66 | 17.3 | 3.97 | 1.0 |
| ETTh1 336 | 83.5% | 64.7% | 0.0% | **3.12** | **1.47** | 1.0 | - | - | 1.0 | 3.72 | 1.5 | 0.94 | - | - | 1.0 | 21.64 | 3.83 | 0.99 | - | 9.03 | 1.0 | 0.58 | 0.55 | 0.64 | 18.95 | 4.16 | 1.0 |
| ETTh1 720 | 74.3% | 52.1% | 0.0% | **3.77** | **1.72** | 1.0 | - | - | 1.0 | 4.35 | 1.63 | 0.92 | - | - | 1.0 | 25.39 | 4.11 | 0.99 | - | 10.35 | 1.0 | 0.68 | 0.61 | 0.6 | 14.67 | 3.59 | 1.0 |
| ETTh2 96 | 19.9% | 9.7% | 0.0% | **2.34** | **1.3** | 1.0 | - | - | 1.0 | 0.97 | 0.75 | 0.95 | - | 21.22 | 1.0 | 2.92 | 1.44 | 1.0 | 25.31 | 4.37 | 1.0 | 0.24 | 0.36 | 0.76 | 38.13 | 5.94 | 1.0 |
| ETTh2 192 | 28.8% | 15.7% | 0.0% | **2.35** | **1.29** | 1.0 | - | - | 1.0 | 0.89 | 0.73 | 0.93 | - | - | 1.0 | 3.3 | 1.53 | 1.0 | 39.41 | 5.41 | 1.0 | 0.27 | 0.38 | 0.74 | 38.82 | 6.04 | 1.0 |
| ETTh2 336 | 91.6% | 74.0% | 0.0% | **3.22** | **1.57** | 1.0 | - | - | 1.0 | 0.86 | 0.7 | 0.91 | - | - | 1.0 | 3.64 | 1.62 | 0.99 | 59.13 | 6.62 | 1.0 | 0.29 | 0.39 | 0.73 | 38.14 | 6.03 | 1.0 |
| ETTh2 720 | 87.1% | 67.1% | 0.0% | **3.36** | **1.62** | 1.0 | - | - | 1.0 | 0.95 | 0.75 | 0.91 | - | - | 1.0 | 4.41 | 1.79 | 0.99 | 84.73 | 7.97 | 1.0 | 0.34 | 0.43 | 0.71 | 26.06 | 4.93 | 1.0 |
| ETTm1 96 | 85.8% | 62.9% | 0.0% | **2.57** | **1.4** | 1.0 | - | - | 1.0 | 20.23 | 3.77 | 1.0 | - | - | 1.0 | 71.87 | 6.81 | 1.0 | - | 36.8 | 1.0 | 0.82 | 0.73 | 0.9 | 18.11 | 4.09 | 1.0 |
| ETTm1 192 | 83.8% | 62.9% | 0.0% | **2.88** | **1.5** | 1.0 | - | - | 1.0 | 24.3 | 4.07 | 1.0 | - | - | 1.0 | 83.69 | 7.35 | 1.0 | - | 46.74 | 1.0 | 0.9 | 0.75 | 0.89 | 17.75 | 4.04 | 1.0 |
| ETTm1 336 | 81.1% | 57.9% | 0.0% | **3.32** | **1.64** | 1.0 | - | - | 1.0 | 21.7 | 3.9 | 1.0 | - | - | 1.0 | 99.42 | 8.02 | 1.0 | - | 56.56 | 1.0 | 0.97 | 0.77 | 0.88 | 17.6 | 4.03 | 1.0 |
| ETTm1 720 | 78.7% | 56.4% | 0.0% | **3.5** | **1.68** | 1.0 | - | - | 1.0 | 23.16 | 4.04 | 0.99 | - | - | 1.0 | - | 8.82 | 1.0 | - | 68.32 | 1.0 | 1.05 | 0.8 | 0.86 | 16.46 | 3.85 | 1.0 |
| ETTm2 96 | 91.4% | 70.9% | 0.0% | **1.21** | **0.94** | 1.0 | - | - | 1.0 | 3.21 | 1.48 | 0.99 | - | - | 1.0 | 14.08 | 3.23 | 1.0 | - | 21.38 | 1.0 | 0.57 | 0.67 | 0.97 | 29.94 | 5.24 | 1.0 |
| ETTm2 192 | 80.2% | 54.9% | 0.0% | **3.43** | **1.62** | 1.0 | - | - | 1.0 | 3.45 | 1.58 | 0.99 | - | - | 1.0 | 17.32 | 3.59 | 1.0 | - | 26.25 | 1.0 | 0.67 | 0.73 | 0.97 | 31.79 | 5.39 | 1.0 |
| ETTm2 336 | 83.1% | 58.2% | 0.0% | **3.42** | **1.62** | 1.0 | - | - | 1.0 | 4.18 | 1.7 | 0.99 | - | - | 1.0 | 20.26 | 3.88 | 1.0 | - | 33.11 | 1.0 | 0.74 | 0.76 | 0.96 | 42.99 | 6.26 | 1.0 |
| ETTm2 720 | 84.4% | 60.0% | 0.0% | **3.55** | **1.64** | 1.0 | - | - | 1.0 | 4.77 | 1.82 | 0.98 | - | - | 1.0 | 22.73 | 4.1 | 1.0 | - | 47.49 | 1.0 | 0.84 | 0.8 | 0.95 | 39.67 | 6.04 | 1.0 |
| Exchange 96 | 69.1% | 41.5% | 0.0% | **1.76** | **1.2** | 1.0 | - | - | 1.0 | 0.26 | 0.37 | 0.81 | - | 33.75 | 1.0 | 0.88 | 0.74 | 0.98 | 6.21 | 2.05 | 1.0 | 0.11 | 0.24 | 0.65 | 5.69 | 2.32 | 1.0 |
| Exchange 192 | 75.8% | 54.6% | 0.0% | **1.71** | **1.14** | 1.0 | - | - | 1.0 | 0.34 | 0.42 | 0.75 | - | - | 1.0 | 1.17 | 0.85 | 0.97 | 9.15 | 2.51 | 1.0 | 0.18 | 0.32 | 0.59 | 7.06 | 2.56 | 1.0 |
| Exchange 336 | 79.4% | 57.3% | 0.0% | **1.85** | **1.22** | 1.0 | - | - | 1.0 | 0.66 | 0.61 | 0.84 | - | - | 1.0 | 1.6 | 1.01 | 0.96 | 15.1 | 3.19 | 1.0 | 0.3 | 0.41 | 0.52 | 8.96 | 2.86 | 1.0 |
| Exchange 720 | 77.7% | 56.0% | 0.0% | **3.01** | **1.53** | 1.0 | - | - | 1.0 | 1.27 | 0.87 | 0.85 | - | - | 1.0 | 2.68 | 1.33 | 0.97 | 30.52 | 4.5 | 1.0 | 1.05 | 0.77 | 0.34 | 13.51 | 3.48 | 1.0 |
| Weather 96 | 95.7% | 85.8% | 0.0% | **30.9** | **2.37** | 1.0 | - | - | 1.0 | 2.76 | 1.05 | 0.96 | - | - | 1.0 | 18.09 | 2.66 | 0.99 | - | 16.73 | 1.0 | 0.5 | 0.58 | 0.94 | - | 30.47 | 1.0 |
| Weather 192 | 89.1% | 78.3% | 0.0% | **84.18** | **3.77** | 1.0 | - | - | 1.0 | 3.21 | 1.13 | 0.94 | - | - | 1.0 | 21.46 | 2.88 | 0.99 | - | 17.38 | 1.0 | 0.57 | 0.62 | 0.93 | - | 30.71 | 1.0 |
| Weather 336 | 89.6% | 78.5% | 0.0% | **84.44** | **3.84** | 1.0 | - | - | 1.0 | 3.72 | 1.22 | 0.93 | - | - | 1.0 | 25.97 | 3.17 | 0.99 | - | 17.87 | 1.0 | 0.64 | 0.66 | 0.92 | - | 32.72 | 1.0 |
| Weather 720 | 89.6% | 78.3% | 0.0% | **86.06** | **3.95** | 1.0 | - | - | 1.0 | 4.73 | 1.3 | 0.92 | - | - | 1.0 | 29.49 | 3.42 | 0.99 | - | 18.24 | 1.0 | 0.74 | 0.71 | 0.91 | - | 34.61 | 1.0 |
| ILI 96 | 98.2% | 85.9% | 0.0% | **30.62** | **4.76** | 1.0 | - | 33.79 | 1.0 | 6.03 | 1.83 | 0.6 | - | 34.41 | 1.0 | 7.69 | 2.02 | 0.7 | - | 8.61 | 0.99 | 4.79 | 1.66 | 0.49 | 29.72 | 4.75 | 0.96 |
| ILI 192 | 98.0% | 85.9% | 0.0% | **31.6** | **4.84** | 1.0 | - | 66.32 | 1.0 | 6.81 | 1.99 | 0.61 | - | 34.36 | 1.0 | 7.06 | 1.93 | 0.68 | - | 8.53 | 0.99 | 4.82 | 1.66 | 0.48 | 31.62 | 4.97 | 0.97 |
| ILI 336 | 98.1% | 86.3% | 0.0% | **38.09** | **5.07** | 1.0 | - | 76.31 | 1.0 | 5.73 | 1.72 | 0.59 | - | 37.13 | 1.0 | 4.49 | 1.55 | 0.63 | 64.81 | 6.48 | 0.99 | 4.11 | 1.47 | 0.46 | 26.03 | 4.48 | 0.97 |
| ILI 720 | 96.7% | 82.3% | 0.0% | **39.14** | **5.23** | 1.0 | - | 29.63 | 1.0 | 4.75 | 1.69 | 0.57 | - | 56.89 | 1.0 | 3.71 | 1.44 | 0.58 | 52.08 | 5.91 | 0.98 | 4.23 | 1.48 | 0.44 | 22.65 | 4.18 | 0.96 |
| CPU 96 | 89.8% | 68.3% | 0.0% | **2.0** | **1.29** | 1.0 | - | 11.58 | 1.0 | 9.88 | 2.85 | 0.99 | - | 28.22 | 1.0 | 19.61 | 4.07 | 1.0 | 97.41 | 8.83 | 1.0 | 0.86 | 0.73 | 0.74 | 28.29 | 5.21 | 1.0 |
| CPU 192 | 92.2% | 71.7% | 0.0% | **2.13** | **1.34** | 1.0 | - | 16.85 | 1.0 | 11.28 | 3.0 | 0.97 | - | 55.6 | 1.0 | 27.35 | 4.74 | 1.0 | - | 13.66 | 1.0 | 0.98 | 0.8 | 0.71 | 29.04 | 5.26 | 1.0 |
| CPU 336 | 92.3% | 73.5% | 0.0% | **2.24** | **1.38** | 1.0 | - | 19.11 | 1.0 | 13.16 | 3.2 | 0.97 | - | 85.21 | 1.0 | 33.27 | 5.21 | 1.0 | - | 15.59 | 1.0 | 1.04 | 0.83 | 0.67 | 28.91 | 5.25 | 1.0 |
| CPU 720 | 93.8% | 71.3% | 0.0% | **2.53** | **1.62** | 1.0 | - | 20.85 | 1.0 | 12.59 | 2.99 | 0.93 | - | 46.05 | 1.0 | 40.63 | 5.65 | 1.0 | - | 17.18 | 1.0 | 0.93 | 0.77 | 0.57 | - | 11.6 | 1.0 |
| Ratio | - | | | 100% | | | 100% | | | 9.38% | | | 100% | | | 50% | | | 87.5% | | | 0% | | | 87.5% | | |
| Count | - | | | 96 | | | 34 | | | 4 | | | 34 | | | 37 | | | 34 | | | 0 | | | 49 | | |

## D.6 RESULTS WITH DIFFERENT HISTORICAL SEQUENCE LENGTH

In this section, we further explore the model's adaptability to varying historical sequence lengths $H$ while keeping the future sequence length $P$ fixed. Specifically, we evaluated the performance when the historical length $H$ takes values from the set $\{192, 336, 720\}$, comparing these results with those of existing baseline models. The experimental outcomes demonstrate that across all tested historical lengths, our model iQR significantly outperforms other models in terms of predictive accuracy. This finding further substantiates the robustness of iQR when dealing with data of differing historical

Table 6: Resource scheduling results on eight datasets with historical length $H = 96$ and prediction length $P \in \{96, 192, 336, 720\}$. The historical and prediction length are set to $H = 36$ and $P \in \{24, 36, 48, 60\}$ for ILI. The quantile is set to 0.95. If the QRE in the result is less than the quantile, it will not be used for comparison. The ratio presents the percentage by which prediction exceeds ground truth. The best result are highlighted in **red bold**, and the second best results are marked with a blue underline.

| Methods | | Ours | | | TimeMixer | | | TimesNet | | | Basisformer | | | iTransformer | | | PatchTST | | | Dlinear | | | FEDformer | | |
|---|---|---|---|---|---|---|---|---|---|---|---|---|---|---|---|---|---|---|---|---|---|---|---|---|---|
| Metrics | | PMSE | PMAE | QRE | PMSE | PMAE | QRE | PMSE | PMAE | QRE | PMSE | PMAE | QRE | PMSE | PMAE | QRE | PMSE | PMAE | QRE | PMSE | PMAE | QRE | PMSE | PMAE | QRE |
| ETTh1 | 96 | 1.441 | 0.939 | 0.953 | 1.727 | 1.025 | 0.938 | 2.673 | 1.256 | 0.938 | 1.655 | 0.984 | 0.949 | 1.94 | 1.114 | 0.943 | 1.644 | 0.992 | 0.929 | 0.514 | 0.513 | 0.678 | 1.433 | 0.927 | 0.867 |
| | 192 | 1.644 | 1.045 | 0.955 | 2.052 | 1.137 | 0.938 | 2.939 | 1.327 | 0.928 | 1.814 | 1.036 | 0.948 | 2.359 | 1.237 | 0.941 | 2.073 | 1.14 | 0.933 | 0.573 | 0.548 | 0.662 | 1.789 | 1.055 | 0.889 |
| | 336 | 1.915 | 1.061 | 0.957 | 2.769 | 1.309 | 0.938 | 2.807 | 1.28 | 0.907 | 1.963 | 1.113 | 0.954 | 2.848 | 1.366 | 0.939 | 2.618 | 1.304 | 0.93 | 0.631 | 0.579 | 0.641 | 2.484 | 1.273 | 0.926 |
| | 720 | 2.231 | 1.17 | 0.95 | 3.693 | 1.505 | 0.919 | 3.157 | 1.362 | 0.87 | 2.001 | 1.106 | 0.952 | 3.775 | 1.579 | 0.917 | 2.865 | 1.344 | 0.896 | 0.755 | 0.646 | 0.605 | 2.705 | 1.341 | 0.914 |
| ETTh2 | 96 | 0.982 | 0.785 | 0.954 | 0.608 | 0.584 | 0.933 | 0.754 | 0.664 | 0.921 | 1.011 | 0.737 | 0.943 | 0.691 | 0.641 | 0.941 | 0.686 | 0.635 | 0.941 | 0.267 | 0.385 | 0.765 | 0.699 | 0.688 | 0.944 |
| | 192 | 1.197 | 0.864 | 0.965 | 0.769 | 0.659 | 0.933 | 0.754 | 0.667 | 0.909 | 1.246 | 0.801 | 0.928 | 0.826 | 0.7 | 0.938 | 0.691 | 0.625 | 0.921 | 0.306 | 0.409 | 0.745 | 0.93 | 0.829 | 0.959 |
| | 336 | 1.053 | 0.806 | 0.953 | 0.942 | 0.757 | 0.943 | 0.785 | 0.676 | 0.893 | 1.217 | 0.817 | 0.901 | 0.892 | 0.739 | 0.93 | 0.79 | 0.675 | 0.915 | 0.317 | 0.418 | 0.732 | 0.989 | 0.849 | 0.949 |
| | 720 | 1.512 | 1.044 | 0.99 | 1.269 | 0.812 | 0.923 | 0.889 | 0.729 | 0.894 | 1.349 | 0.851 | 0.924 | 1.088 | 0.831 | 0.934 | 1.001 | 0.787 | 0.931 | 0.379 | 0.461 | 0.722 | 2.058 | 1.217 | 0.951 |
| ETTm1 | 96 | 1.855 | 1.118 | 0.951 | 1.301 | 0.86 | 0.925 | 1.312 | 0.865 | 0.893 | 1.583 | 0.92 | 0.942 | 1.372 | 0.904 | 0.928 | 1.462 | 0.923 | 0.925 | 0.822 | 0.733 | 0.892 | 1.201 | 0.847 | 0.905 |
| | 192 | 1.62 | 1.026 | 0.951 | 1.508 | 0.929 | 0.932 | 1.523 | 0.947 | 0.909 | 1.892 | 1.033 | 0.95 | 1.663 | 1.014 | 0.93 | 1.701 | 1.006 | 0.925 | 0.92 | 0.766 | 0.885 | 1.42 | 0.931 | 0.908 |
| | 336 | 1.887 | 1.128 | 0.955 | 1.625 | 0.977 | 0.922 | 1.812 | 1.051 | 0.915 | 2.065 | 1.087 | 0.948 | 1.93 | 1.1 | 0.927 | 1.907 | 1.069 | 0.918 | 1.029 | 0.805 | 0.876 | 1.693 | 1.034 | 0.922 |
| | 720 | 1.957 | 1.156 | 0.956 | 2.23 | 1.168 | 0.93 | 2.379 | 1.252 | 0.93 | 2.419 | 1.204 | 0.949 | 2.465 | 1.263 | 0.926 | 2.416 | 1.235 | 0.925 | 1.144 | 0.85 | 0.862 | 1.876 | 1.089 | 0.893 |
| ETTm2 | 96 | 0.595 | 0.629 | 0.967 | 0.379 | 0.459 | 0.92 | 0.414 | 0.493 | 0.93 | 0.636 | 0.592 | 0.946 | 0.44 | 0.523 | 0.939 | 0.358 | 0.458 | 0.931 | 0.449 | 0.589 | 0.955 | 0.443 | 0.53 | 0.958 |
| | 192 | 0.729 | 0.682 | 0.96 | 0.59 | 0.563 | 0.93 | 0.536 | 0.557 | 0.926 | 0.852 | 0.681 | 0.939 | 0.586 | 0.601 | 0.935 | 0.512 | 0.543 | 0.931 | 0.565 | 0.661 | 0.955 | 0.66 | 0.647 | 0.96 |
| | 336 | 0.825 | 0.721 | 0.961 | 0.919 | 0.673 | 0.925 | 0.692 | 0.636 | 0.926 | 1.279 | 0.841 | 0.943 | 0.753 | 0.679 | 0.933 | 0.698 | 0.636 | 0.936 | 0.666 | 0.716 | 0.954 | 0.788 | 0.73 | 0.968 |
| | 720 | 0.785 | 0.715 | 0.982 | 1.134 | 0.83 | 0.947 | 0.926 | 0.728 | 0.916 | 1.51 | 0.912 | 0.929 | 1.09 | 0.822 | 0.937 | 0.917 | 0.726 | 0.925 | 0.795 | 0.78 | 0.947 | 1.023 | 0.86 | 0.967 |
| Exchange | 96 | 0.29 | 0.402 | 0.956 | 0.449 | 0.525 | 0.969 | 0.285 | 0.407 | 0.805 | 0.493 | 0.532 | 0.973 | 0.377 | 0.487 | 0.96 | 0.48 | 0.53 | 0.968 | 0.128 | 0.268 | 0.652 | 0.878 | 0.799 | 0.994 |
| | 192 | 0.422 | 0.511 | 0.977 | 0.824 | 0.697 | 0.944 | 0.421 | 0.494 | 0.739 | 0.847 | 0.703 | 0.966 | 0.688 | 0.657 | 0.951 | 0.865 | 0.718 | 0.965 | 0.225 | 0.353 | 0.596 | 1.359 | 0.999 | 0.996 |
| | 336 | 0.519 | 0.58 | 0.956 | 1.998 | 1.128 | 0.983 | 0.767 | 0.691 | 0.837 | 2.001 | 1.097 | 0.979 | 1.201 | 0.879 | 0.951 | 1.643 | 0.996 | 0.968 | 0.312 | 0.422 | 0.524 | 2.361 | 1.331 | 0.999 |
| | 720 | 0.822 | 0.754 | 0.953 | 6.254 | 1.961 | 0.981 | 1.494 | 0.994 | 0.852 | 7.001 | 2.089 | 0.994 | 2.751 | 1.357 | 0.97 | 4.874 | 1.779 | 0.99 | 0.462 | 0.545 | 0.353 | 5.28 | 1.974 | 0.999 |
| Weather | 96 | 1.079 | 0.789 | 0.966 | 0.595 | 0.442 | 0.889 | 0.628 | 0.486 | 0.857 | 0.911 | 0.512 | 0.912 | 0.909 | 0.528 | 0.905 | 0.779 | 0.489 | 0.913 | 0.454 | 0.571 | 0.934 | 1.377 | 0.898 | 0.856 |
| | 192 | 1.289 | 0.884 | 0.951 | 0.786 | 0.566 | 0.907 | 0.74 | 0.541 | 0.835 | 1.136 | 0.597 | 0.898 | 1.25 | 0.612 | 0.899 | 1.021 | 0.586 | 0.901 | 0.533 | 0.621 | 0.925 | 1.436 | 0.923 | 0.845 |
| | 336 | 1.709 | 1.075 | 0.958 | 0.86 | 0.602 | 0.882 | 0.906 | 0.62 | 0.839 | 1.475 | 0.704 | 0.897 | 1.466 | 0.729 | 0.887 | 1.338 | 0.71 | 0.907 | 0.624 | 0.673 | 0.917 | 1.664 | 0.999 | 0.906 |
| | 720 | 2.729 | 1.392 | 0.952 | 1.212 | 0.742 | 0.899 | 1.027 | 0.691 | 0.812 | 1.861 | 0.813 | 0.898 | 1.771 | 0.839 | 0.888 | 1.794 | 0.847 | 0.905 | 0.744 | 0.735 | 0.906 | 2.43 | 1.27 | 0.934 |
| ILI | 24 | 33.407 | 4.943 | 0.967 | 18.574 | 3.395 | 0.876 | 3.859 | 1.641 | 0.566 | 10.712 | 2.592 | 0.881 | 3.159 | 1.396 | 0.567 | 10.857 | 2.513 | 0.877 | 1.49 | 1.01 | 0.44 | 6.417 | 2.135 | 0.77 |
| | 36 | 54.821 | 6.386 | 0.956 | 27.936 | 3.845 | 0.923 | 5.7 | 1.847 | 0.603 | 15.209 | 3.09 | 0.89 | 4.654 | 1.618 | 0.619 | 12.832 | 2.628 | 0.902 | 2.107 | 1.148 | 0.461 | 7.18 | 2.216 | 0.793 |
| | 48 | 97.343 | 8.349 | 0.95 | 21.284 | 3.639 | 0.898 | 6.217 | 1.829 | 0.592 | 15.746 | 3.175 | 0.9 | 7.857 | 2.062 | 0.676 | 20.766 | 3.135 | 0.914 | 2.722 | 1.382 | 0.477 | 9.23 | 2.477 | 0.824 |
| | 60 | 95.065 | 8.22 | 0.95 | 27.673 | 3.62 | 0.923 | 7.231 | 2.13 | 0.61 | 19.6 | 3.404 | 0.923 | 8.204 | 2.104 | 0.685 | 21.469 | 3.186 | 0.911 | 2.743 | 1.388 | 0.487 | 9.876 | 2.557 | 0.829 |
| CPU | 96 | 2.766 | 1.512 | 0.952 | 1.827 | 1.007 | 0.929 | 2.832 | 1.432 | 0.911 | 3.483 | 1.551 | 0.936 | 2.21 | 1.265 | 0.889 | 2.694 | 1.369 | 0.914 | 0.768 | 0.714 | 0.741 | 1.78 | 1.085 | 0.846 |
| | 192 | 2.917 | 1.517 | 0.953 | 2.259 | 1.153 | 0.916 | 4.417 | 1.837 | 0.922 | 4.042 | 1.692 | 0.915 | 2.318 | 1.296 | 0.856 | 3.597 | 1.631 | 0.91 | 0.801 | 0.744 | 0.714 | 2.379 | 1.279 | 0.876 |
| | 336 | 3.096 | 1.548 | 0.955 | 2.825 | 1.314 | 0.926 | 5.29 | 2.034 | 0.937 | 4.208 | 1.683 | 0.895 | 2.129 | 1.239 | 0.804 | 3.649 | 1.632 | 0.888 | 0.84 | 0.763 | 0.672 | 2.008 | 1.164 | 0.839 |
| | 720 | 2.484 | 1.387 | 0.981 | 3.343 | 1.493 | 0.916 | 10.061 | 2.735 | 0.886 | 5.906 | 1.939 | 0.907 | 3.041 | 1.45 | 0.762 | 4.705 | 1.817 | 0.886 | 0.635 | 0.667 | 0.569 | 2.396 | 1.335 | 0.834 |
| Ratio | | 100% | | | 9.375% | | | 0% | | | 21.875% | | | 12.5% | | | 12.5% | | | 9.375% | | | 31.25% | | |
| Count | | 88 | | | 1 | | | 0 | | | 11 | | | 8 | | | 0 | | | 7 | | | 21 | | |

Table 7: Resource scheduling results on eight datasets with historical length $H = 96$ and prediction length $P \in \{96, 192, 336, 720\}$. The historical and prediction length are set to $H = 36$ and $P \in \{24, 36, 48, 60\}$ for ILI. The quantile is set to 0.9. If the QRE in the result is less than the quantile, it will not be used for comparison. The best result are highlighted in **red bold**, and the second best results are marked with a blue underline.

| Methods | | Ours | | | TimeMixer | | | TimesNet | | | Basisformer | | | iTransformer | | | PatchTST | | | Dlinear | | | FEDformer | | |
|---|---|---|---|---|---|---|---|---|---|---|---|---|---|---|---|---|---|---|---|---|---|---|---|---|---|
| Metrics | | PMSE | PMAE | QRE | PMSE | PMAE | QRE | PMSE | PMAE | QRE | PMSE | PMAE | QRE | PMSE | PMAE | QRE | PMSE | PMAE | QRE | PMSE | PMAE | QRE | PMSE | PMAE | QRE |
| ETTh1 | 96 | 1.185 | 0.782 | 0.904 | 1.209 | 0.81 | 0.884 | 1.514 | 0.951 | 0.884 | 1.23 | 0.82 | 0.901 | 1.287 | 0.874 | 0.887 | 1.187 | 0.801 | 0.874 | 0.57 | 0.546 | 0.66 | 1.106 | 0.764 | 0.771 |
| | 192 | 1.239 | 0.842 | 0.904 | 1.345 | 0.865 | 0.872 | 1.705 | 1.015 | 0.885 | 1.311 | 0.847 | 0.891 | 1.53 | 0.963 | 0.887 | 1.433 | 0.901 | 0.876 | 0.629 | 0.577 | 0.639 | 1.244 | 0.833 | 0.793 |
| | 336 | 1.821 | 1.027 | 0.9 | 1.547 | 0.943 | 0.875 | 2.308 | 1.131 | 0.849 | 1.347 | 0.868 | 0.884 | 1.816 | 1.052 | 0.883 | 1.674 | 0.991 | 0.864 | 0.753 | 0.645 | 0.604 | 1.531 | 0.969 | 0.866 |
| | 720 | 1.798 | 1.032 | 0.902 | 1.962 | 1.096 | 0.852 | 2.515 | 1.176 | 0.786 | 1.453 | 0.909 | 0.885 | 2.303 | 1.194 | 0.846 | 1.881 | 1.055 | 0.827 | 0.511 | 0.51 | 0.511 | 1.88 | 1.085 | 0.852 |
| ETTh2 | 96 | 0.924 | 0.734 | 0.955 | 0.697 | 0.643 | 0.82 | 0.648 | 0.62 | 0.831 | 0.782 | 0.629 | 0.899 | 0.49 | 0.519 | 0.889 | 0.421 | 0.474 | 0.872 | 0.301 | 0.405 | 0.741 | 0.417 | 0.502 | 0.878 |
| | 192 | 1.466 | 1.007 | 0.949 | 0.885 | 0.751 | 0.846 | 0.753 | 0.682 | 0.82 | 0.872 | 0.668 | 0.884 | 0.578 | 0.563 | 0.883 | 0.54 | 0.534 | 0.873 | 0.314 | 0.415 | 0.729 | 0.639 | 0.645 | 0.888 |
| | 336 | 1.982 | 1.184 | 0.961 | 0.894 | 0.741 | 0.786 | 0.859 | 0.734 | 0.822 | 1.024 | 0.734 | 0.888 | 0.618 | 0.589 | 0.876 | 0.516 | 0.521 | 0.838 | 0.38 | 0.462 | 0.723 | 0.564 | 0.583 | 0.892 |
| | 720 | 2.132 | 1.228 | 0.967 | 0.863 | 0.761 | 0.804 | 1.091 | 0.873 | 0.845 | 0.99 | 0.75 | 0.871 | 0.686 | 0.629 | 0.871 | 0.704 | 0.662 | 0.892 | 0.262 | 0.379 | 0.76 | 0.732 | 0.684 | 0.895 |
| ETTm1 | 96 | 2.282 | 1.173 | 0.901 | 0.885 | 0.669 | 0.851 | 0.915 | 0.684 | 0.808 | 1.111 | 0.743 | 0.875 | 0.929 | 0.708 | 0.864 | 0.993 | 0.727 | 0.866 | 0.791 | 0.692 | 0.866 | 0.845 | 0.66 | 0.844 |
| | 192 | 1.895 | 1.079 | 0.907 | 1.078 | 0.745 | 0.866 | 1.024 | 0.738 | 0.832 | 1.285 | 0.815 | 0.89 | 1.116 | 0.787 | 0.868 | 1.077 | 0.748 | 0.842 | 0.914 | 0.741 | 0.854 | 0.961 | 0.711 | 0.845 |
| | 336 | 1.89 | 1.085 | 0.907 | 1.176 | 0.783 | 0.856 | 1.289 | 0.846 | 0.861 | 1.521 | 0.903 | 0.903 | 1.289 | 0.854 | 0.868 | 1.318 | 0.848 | 0.856 | 1.045 | 0.798 | 0.843 | 1.096 | 0.766 | 0.846 |
| | 720 | 2.203 | 1.168 | 0.907 | 1.461 | 0.902 | 0.856 | 1.594 | 0.97 | 0.87 | 1.509 | 0.926 | 0.886 | 1.563 | 0.965 | 0.868 | 1.556 | 0.937 | 0.85 | 0.686 | 0.651 | 0.85 | 1.222 | 0.812 | 0.82 |
| ETTm2 | 96 | 0.579 | 0.58 | 0.927 | 0.256 | 0.365 | 0.852 | 0.294 | 0.395 | 0.87 | 0.409 | 0.454 | 0.9 | 0.299 | 0.413 | 0.888 | 0.264 | 0.365 | 0.852 | 0.397 | 0.534 | 0.881 | 0.905 | 0.667 | 0.771 |
| | 192 | 1.138 | 0.844 | 0.935 | 0.425 | 0.472 | 0.887 | 0.363 | 0.445 | 0.873 | 0.561 | 0.536 | 0.9 | 0.39 | 0.471 | 0.883 | 0.344 | 0.421 | 0.856 | 0.492 | 0.595 | 0.893 | 1.037 | 0.734 | 0.793 |
| | 336 | 1.19 | 0.889 | 0.94 | 0.524 | 0.509 | 0.864 | 0.481 | 0.5 | 0.863 | 0.763 | 0.624 | 0.901 | 0.497 | 0.529 | 0.881 | 0.457 | 0.494 | 0.877 | 0.629 | 0.675 | 0.891 | 1.361 | 0.888 | 0.866 |
| | 720 | 1.517 | 1.036 | 0.959 | 0.619 | 0.572 | 0.861 | 0.667 | 0.616 | 0.87 | 0.986 | 0.707 | 0.886 | 0.682 | 0.626 | 0.886 | 0.569 | 0.536 | 0.85 | 0.302 | 0.463 | 0.887 | 1.643 | 0.983 | 0.852 |
| Exchange | 96 | 0.199 | 0.315 | 0.9 | 0.282 | 0.405 | 0.917 | 0.359 | 0.454 | 0.872 | 0.6 | 0.596 | 0.949 | 0.318 | 0.458 | 0.917 | 0.306 | 0.447 | 0.907 | 0.225 | 0.352 | 0.598 | 0.66 | 0.698 | 0.99 |
| | 192 | 0.396 | 0.486 | 0.91 | 0.673 | 0.624 | 0.906 | 0.632 | 0.611 | 0.873 | 1.077 | 0.81 | 0.957 | 0.517 | 0.561 | 0.916 | 0.627 | 0.641 | 0.93 | 0.313 | 0.422 | 0.53 | 1.005 | 0.843 | 0.989 |
| | 336 | 1.005 | 0.801 | 0.914 | 1.099 | 0.812 | 0.932 | 0.735 | 0.673 | 0.83 | 1.91 | 1.096 | 0.978 | 0.91 | 0.757 | 0.925 | 1.2 | 0.894 | 0.928 | 0.464 | 0.546 | 0.563 | 1.656 | 1.066 | 0.994 |
| | 720 | 0.979 | 0.804 | 0.913 | 3.889 | 1.569 | 0.965 | 1.494 | 0.996 | 0.862 | 4.357 | 1.675 | 0.978 | 2.38 | 1.257 | 0.961 | 3.889 | 1.569 | 0.965 | 0.127 | 0.266 | 0.562 | 4.216 | 1.735 | 0.998 |
| Weather | 96 | 0.767 | 0.58 | 0.905 | 0.341 | 0.343 | 0.83 | 0.331 | 0.338 | 0.817 | 0.535 | 0.404 | 0.832 | 0.47 | 0.361 | 0.838 | 0.463 | 0.375 | 0.864 | 0.473 | 0.572 | 0.904 | 0.421 | 0.41 | 0.894 |
| | 192 | 0.983 | 0.742 | 0.92 | 0.416 | 0.408 | 0.837 | 0.428 | 0.426 | 0.725 | 0.633 | 0.458 | 0.836 | 0.602 | 0.45 | 0.835 | 0.586 | 0.452 | 0.855 | 0.573 | 0.633 | 0.897 | 0.564 | 0.502 | 0.859 |
| | 336 | 1.093 | 0.805 | 0.904 | 0.551 | 0.469 | 0.825 | 0.579 | 0.493 | 0.708 | 0.824 | 0.543 | 0.83 | 0.894 | 0.553 | 0.835 | 0.695 | 0.493 | 0.825 | 0.697 | 0.704 | 0.899 | 0.692 | 0.559 | 0.854 |
| | 720 | 1.506 | 0.981 | 0.909 | 0.692 | 0.546 | 0.833 | 0.71 | 0.582 | 0.737 | 1.228 | 0.672 | 0.839 | 1.032 | 0.619 | 0.831 | 1.009 | 0.63 | 0.844 | 0.688 | 0.71 | 0.921 | 0.94 | 0.681 | 0.827 |
| ILI | 24 | 2.465 | 1.374 | 0.905 | 13.339 | 2.583 | 0.805 | 9.814 | 1.997 | 0.807 | 5.084 | 1.57 | 0.818 | 6.995 | 1.874 | 0.658 | 8.0 | 1.755 | 0.825 | 2.735 | 1.385 | 0.486 | 6.898 | 2.097 | 0.753 |
| | 36 | 6.761 | 2.291 | 0.901 | 9.662 | 2.148 | 0.807 | 9.826 | 1.958 | 0.776 | 3.526 | 1.432 | 0.817 | 4.949 | 1.614 | 0.623 | 5.941 | 1.558 | 0.789 | 2.719 | 1.381 | 0.477 | 5.785 | 1.904 | 0.741 |
| | 48 | 34.879 | 4.981 | 0.902 | 12.691 | 2.782 | 0.779 | 7.785 | 1.771 | 0.837 | 2.658 | 1.232 | 0.742 | 4.813 | 1.602 | 0.633 | 6.684 | 1.765 | 0.824 | 2.106 | 1.147 | 0.461 | 5.431 | 1.829 | 0.729 |
| | 60 | 36.857 | 5.099 | 0.912 | 7.125 | 1.937 | 0.789 | 6.154 | 1.701 | 0.764 | 4.111 | 1.516 | 0.791 | 2.79 | 1.297 | 0.553 | 4.992 | 1.512 | 0.775 | 1.488 | 1.01 | 0.44 | 4.144 | 1.705 | 0.673 |
| CPU | 96 | 1.091 | 0.908 | 0.906 | 1.382 | 0.985 | 0.849 | 1.84 | 1.136 | 0.877 | 1.963 | 1.16 | 0.885 | 1.14 | 0.861 | 0.824 | 1.604 | 1.03 | 0.864 | 0.786 | 0.734 | 0.711 | 1.573 | 1.013 | 0.841 |
| | 192 | 1.858 | 1.207 | 0.905 | 2.168 | 1.247 | 0.848 | 2.249 | 1.254 | 0.858 | 2.196 | 1.257 | 0.857 | 1.595 | 1.058 | 0.82 | 2.094 | 1.227 | 0.857 | 0.826 | 0.755 | 0.67 | 1.763 | 1.076 | 0.842 |
| | 336 | 2.891 | 1.512 | 0.907 | 2.101 | 1.238 | 0.812 | 2.788 | 1.435 | 0.863 | 3.009 | 1.412 | 0.836 | 1.795 | 1.119 | 0.779 | 2.242 | 1.244 | 0.81 | 0.626 | 0.662 | 0.567 | 2.123 | 1.222 | 0.825 |
| | 720 | 2.595 | 1.358 | 0.912 | 3.235 | 1.538 | 0.833 | 4.343 | 1.785 | 0.845 | 4.55 | 1.703 | 0.845 | 2.518 | 1.311 | 0.742 | 3.556 | 1.553 | 0.813 | 0.753 | 0.705 | 0.736 | 3.213 | 1.506 | 0.778 |
| Ratio | | 100% | | | 12.5% | | | 0% | | | 28.125% | | | 12.5% | | | 12.5% | | | 6.25% | | | 12.5% | | |
| Count | | 92 | | | 2 | | | 0 | | | 19 | | | 6 | | | 0 | | | 6 | | | 4 | | |

lengths. As depicted in Figure 8, under the condition of a fixed prediction length, iQR achieves higher QRE at each configuration of historical length. Unlike other methods, iQR does not rely on GPU resources, and its training duration is significantly shorter than other approaches, thereby highlighting its performance superiority across various parameter settings.

Table 8: QRE results for resource scheduling on ETTh1 dataset with historical length $H = \{192, 336, 720\}$ and prediction length $P = 96$. The quantile is 0.9. If the QRE in the result is less than the quantile, it will not be used for comparison. The best result are highlighted in **red bold**.

| Methods | H | P | PMSE | PMAE | QRE | Time(s) | GPU(GB) |
|---|---|---|---|---|---|---|---|
| iQR | 192 | 96 | 1.12 | 0.74 | **0.91** | 1.15 | 0 |
| iQR | 336 | 96 | 1.19 | 0.79 | **0.92** | 1.18 | 0 |
| iQR | 720 | 96 | 1.22 | 0.83 | **0.92** | 1.21 | 0 |
| TimeMixer | 192 | 96 | 1.08 | 0.75 | 0.89 | 375.73 | 0.55 |
| TimeMixer | 336 | 96 | 1.01 | 0.72 | 0.87 | 382.28 | 0.94 |
| TimeMixer | 720 | 96 | 1.20 | 0.85 | 0.89 | 433.39 | 2.39 |
| TimesNet | 192 | 96 | 1.26 | 0.83 | 0.88 | 837.19 | 0.19 |
| TimesNet | 336 | 96 | 1.25 | 0.82 | 0.88 | 851.84 | 0.27 |
| TimesNet | 720 | 96 | 1.13 | 0.78 | 0.85 | 889.05 | 0.48 |
| Basisformer | 192 | 96 | 1.29 | 0.86 | 0.89 | 359.55 | 0.04 |
| Basisformer | 336 | 96 | 1.23 | 0.84 | 0.87 | 359.23 | 0.04 |
| Basisformer | 720 | 96 | 1.35 | 0.90 | 0.89 | 348.16 | 0.05 |
| iTransformer | 192 | 96 | 1.12 | 0.80 | 0.89 | 82.10 | 0.01 |
| iTransformer | 336 | 96 | 1.13 | 0.81 | 0.88 | 201.57 | 0.01 |
| iTransformer | 720 | 96 | 1.17 | 0.83 | 0.89 | 197.04 | 0.01 |
| PatchTST | 192 | 96 | 1.05 | 0.74 | 0.88 | 166.41 | 0.41 |
| PatchTST | 336 | 96 | 1.11 | 0.76 | 0.88 | 199.85 | 1.78 |
| PatchTST | 720 | 96 | 1.09 | 0.76 | 0.88 | 223.40 | 2.09 |
| DLinear | 192 | 96 | 0.46 | 0.48 | 0.67 | 107.57 | 0.01 |
| DLinear | 336 | 96 | 0.45 | 0.47 | 0.68 | 119.17 | 0.01 |
| DLinear | 720 | 96 | 0.45 | 0.47 | 0.67 | 127.84 | 0.01 |
| FEDformer | 192 | 96 | 1.17 | 0.82 | 0.87 | 1073.57 | 1.94 |
| FEDformer | 336 | 96 | 1.36 | 0.91 | 0.89 | 1102.01 | 2.23 |
| FEDformer | 720 | 96 | 1.39 | 0.91 | 0.86 | 1137.32 | 3.13 |

## D.7 RESULTS ON BIG DATASETS

To further explore the computational efficiency and scalability of the iQR framework, we paid particular attention to its performance when handling large-scale datasets. For this evaluation, the Bitcoin historical dataset provided on Kaggle is adopted [6], which records Bitcoin transaction data every minute from 2012 to 2021. After preprocessing, the dataset reached a length of 3,330,541 records, involving 7 features.

In this assessment, we compared iQR with traditional time series analysis methods, such as AR and ARIMA, as well as deep learning approaches, including LSTM and Transformer. Notably, to ensure a fair comparison, we modified the traditional methods to a quantile version and introduced a quantile regression loss function for the deep learning models.

The test results in Table 9 demonstrated that iQR outperformed these traditional methods in terms of memory usage and processing speed. Specifically, when processing over three million data records, the training time for LSTM and Transformer was 100 and 300 times that of iQR, respectively. This significant difference highlights iQR's capability to handle large datasets in resource-constrained environments, especially without the need for GPU resource support.

Furthermore, the lightweight nature of iQR allows it to complete training within seconds on CPUs, significantly reducing computational overhead. This increase in efficiency not only showcases the potential in practical applications but also proves its practical value in resource scheduling problems, especially in scenarios that require rapid and accurate predictions.

---

[6] https://www.kaggle.com/datasets/mczielinski/bitcoin-historical-data

iQR framework has shown excellent computational efficiency and good scalability when dealing with large-scale datasets, making it an ideal choice for addressing resource scheduling problems, particularly in real-time application scenarios where computational resources are limited.

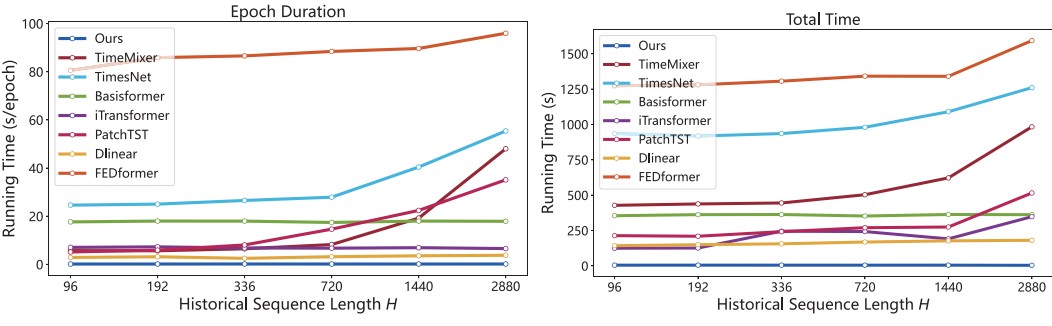

(a) Training epoch duration efficiency analysis.    (b) Total time efficiency analysis.

Figure 6: Additional running efficiency analysis.

Table 9: QRE results for resource scheduling on Bitcoin dataset with historical length $H = 96$ and prediction length $P = 96$. If PMSE or PMAE is greater than 100, the symbol - is used to replace the original value. The quantile is 0.9. If the QRE in the result is less than the quantile, it will not be used for comparison. The best result are highlighted in **red bold**.

| Methods | Length | Time(s) | GPU(GB) | PMSE | PMAE | QRE |
|---|---|---|---|---|---|---|
| iQR | $10^3$ | 0.13 | 0 | 1.55 | 1.04 | **0.93** |
| iQR | $10^4$ | 1.14 | 0 | 1.17 | 0.87 | **0.91** |
| iQR | $10^5$ | 15.21 | 0 | 1.28 | 1.00 | **0.91** |
| iQR | $10^6$ | 101.84 | 0 | 1.49 | 1.29 | **0.90** |
| QAR | $10^3$ | 1.03 | 0 | - | 43.84 | 0.15 |
| QAR | $10^4$ | 2.41 | 0 | - | - | 0.16 |
| QAR | $10^5$ | 19.15 | 0 | - | - | 0.11 |
| QAR | $10^6$ | 166.52 | 0 | - | - | 0.07 |
| QARIMA | $10^3$ | 1.67 | 0 | - | 16.07 | 0.59 |
| QARIMA | $10^4$ | 4.92 | 0 | - | 17.52 | 0.74 |
| QARIMA | $10^5$ | 20.39 | 0 | - | - | 0.38 |
| QARIMA | $10^6$ | 204.45 | 0 | - | - | 0.52 |
| LSTM | $10^3$ | 10.37 | 0.02 | - | 15.83 | 0.69 |
| LSTM | $10^4$ | 101.49 | 0.02 | - | - | 0.50 |
| LSTM | $10^5$ | 1179.03 | 0.02 | - | - | 0.95 |
| LSTM | $10^6$ | 11706.33 | 0.02 | - | - | 0.90 |
| GRU | $10^3$ | 10.5 | 0.02 | - | 29.75 | 0.81 |
| GRU | $10^4$ | 105.73 | 0.02 | - | - | 0.79 |
| GRU | $10^5$ | 1067.28 | 0.02 | - | - | 0.97 |
| GRU | $10^6$ | 11790.11 | 0.02 | - | - | 0.97 |
| Transformer | $10^3$ | 32.15 | 0.08 | - | 11.70 | 0.95 |
| Transformer | $10^4$ | 302.93 | 0.08 | - | - | 0.25 |
| Transformer | $10^5$ | 3147.43 | 0.08 | - | - | 0.98 |
| Transformer | $10^6$ | 31278.46 | 0.08 | - | - | 0.97 |

## D.8 EFFICIENCY ANALYSIS

We further provide a detailed analysis of total time and one-epoch execution time to compare different models. Specifically, the total time of iQR remains relatively stable as the length of historical sequences increases, showing its efficiency in long sequence training. As shown in Figure 6, as the sequence length grows from 96 to 2880, the total time of iQR only increases from 1.082 s to 1.141 s. In addition, we take the single-channel total time in the iQR model as the one-epoch execution time, and the results show that the computational efficiency of iQR is similar within each channel, regardless of the large increase in sequence length.

For TimeMixer and TimesNet, under the same conditions, both the total time and the duration of individual epochs show a significant increase with sequence length, which directly leads to their poor performance in long sequence prediction tasks. Basisformer maintains similar performance across different scale sequences, outperforming other baselines, but the time spent is still higher than our method. Although PatchTST uses time series segmentation to improve learning efficiency, the result shows a significant increase in total time as the sequence length increases, revealing the efficiency limitations of the segmentation strategy for long sequence processing. This further emphasizes the applicability of iQR on data at different scales. Dlinear outperforms the transformer-based model in terms of less memory usage and time spent. FEDformer consistently exhibits the largest resource usage, showing its limitations in efficiency and scalability. In conclusion, iQR provides both efficient and resource-saving solutions for resource scheduling tasks.

## E    PSEUDOCODE

We further provide pseudocode of the proposed algorithm iQR, offering a concise, step-by-step outline of its functionality and logic. The pseudocode can be found in Pseudocode 1.

---

**Algorithm 1:** Business-driven identification prediction framework (iQR)

---

**Input:** Time series $\mathbf{X}$, historical length $H$, prediction length $P$, hyperparameters $\alpha, \beta, \tau, \lambda$
**Output:** Prediction sequence $\hat{\mathbf{Y}}$

---

**for** each channel $i = 1, \ldots, d_x$ **do**

    Split data $X^i$ into a training set $X^i_{train}$ and a test set $X^i_{test}$

    /\* **Phase I: Global Identification**     \*/

    Perform data normalization

    Select the 1st set of basis vectors using FFT on $X^i_{train}$, generating global basis vectors $\boldsymbol{\Theta}_g$

    Select the 2nd set of basis vectors using SOR, resulting in the significant global basis $\hat{\boldsymbol{\Theta}}^i_g$

    Perform $L_1$QR to compute the global identification $\hat{Y}^i_g$

    /\* **Phase II: Local Identification**     \*/

    **for** each test batch $b = 1, \ldots, N$ **do**

        Fuse global and local knowledge, yielding the fused sequence $\hat{X}^i = \mathbf{w}X^i + (1 - \mathbf{w})Y^i_g$

        Select the 1st set of local basis using FFT on $\hat{X}^i_{l_b,t}$, yielding local basis vectors $\boldsymbol{\Theta}^i_{l_b,t}$

        Select the 2nd set of local basis using SOR, obtaining the significant local basis $\hat{\boldsymbol{\Theta}}^i_{l_b,t}$

        Enhance the input sequence using STL, resulting in $\tilde{X}^i_{l_b,t}$

        Perform $L_1$QR to obtain the local identification $\hat{Y}^i_{l_b,t}$

/\* **Phase III: Concatenation**     \*/

Construct the complete prediction sequence $\hat{\mathbf{Y}}$

**return** $\hat{\mathbf{Y}}$

---

## F    THEORETICAL PROOF

### F.1    PROOF OF LEMMA 2

*Proof.* Let $E$ denote the error, $Y$ represent the ground truth, $\mathbf{X}$ be the basis function library, and $\theta$ be the coefficients. Then, we have $E = Y - \mathbf{X}\theta$. Therefore, the optimization function $O(\theta)$ can be written in the following form:

$$O(\theta) = \frac{1}{2}E^\top E = \frac{1}{2}(Y - \mathbf{X}\theta)^\top(Y - \mathbf{X}\theta), \tag{9}$$

$$= \frac{1}{2}\theta^\top \mathbf{X}^\top \mathbf{X}\theta - \mathbf{X}^\top Y\theta + \frac{1}{2}Y^\top Y.$$

Since this is an unbiased least squares estimate, it holds true if and only if the following equation is satisfied

$$\frac{\partial O}{\partial \theta} = \mathbf{X}^\top \mathbf{X} \theta - \mathbf{X}^\top Y = \mathbf{X}^\top (\mathbf{X}\theta - Y) = -\mathbf{X}^\top E = 0. \tag{10}$$

Equation 10 proof the orthogonality principle. □

### F.2 PROOF OF LEMMA 3

*Proof.* The proof of the number of basis vectors is conducted using mathematical induction.

(1) When there is only 1 basis function, it is obviously true.

(2) Assuming that the statement holds true for $k$ basis vectors, we consider the case when there are $(k + 1)$ basis vectors. Since the residual corresponding to the representation of the target using $k$ basis vectors is orthogonal to any linear combination of these $k$ basis vectors (Lemma 2), and the additional $(k + 1)$th basis function is orthogonal to the $k$ basis vectors, the projection of the linear combination of the $k$ basis vectors onto the $(k + 1)$ basis vectors and the residual vector must be zero (i.e., the subsequent selections do not affect the influence of the $(k + 1)$th basis function on the residual). The problem then reduces to the projection process of the $(k + 1)$ basis vectors onto E. According to (1), the first greedy selection is also globally optimal. After the first greedy selection, there remain $k$ basis vectors, and by the assumption, the conclusion holds. □

### F.3 PROOF OF THEOREM 1

*Proof.* Consider the regression model $Y = \mathbf{X}\theta + E$.

According to Corollary 2, the matrix $X$ is decomposed to $\mathbf{X} = \mathbf{VR}$, where $\mathbf{V}_{m \times n}$ is a matrix with pairwise orthogonal columns $\mathbf{V}_i$, and $\mathbf{R}_{n \times n}$ is an upper triangle matrix with diagonal entries equal to 1. Besides, $\mathbf{V}^\top \mathbf{V} = \mathbf{\Lambda} = \mathrm{diag}(d_1, d_2, \ldots, d_n)$ with $d_i = \mathbf{V}_i^\top \mathbf{V}_i$.

Then, the regression model is remarked as

$$Y = \mathbf{V}B + E, \tag{11}$$

in which $\mathbf{R}\theta = B$. The solution to the regression model is calculated by $B = (\mathbf{V}^\top \mathbf{V})^{-1} \mathbf{V}^\top Y = \mathbf{\Lambda}^{-1} \mathbf{V}^\top Y$. The values are denoted as $B_i = (\mathbf{V}_i^\top Y)/(\mathbf{V}_i^\top \mathbf{V}_i)$, $i \in [1, n]$.

According to Lemma 2, $\mathbf{V}_i$ is orthogonal to $\mathbf{V}_j$ and $E$. By multiplying $Y$ itself and dividing by $N$, the output variance is given by

$$\frac{1}{N} Y^\top Y = \frac{1}{N} \sum_{i=1}^{n} B_i^2 \mathbf{V}_i^\top \mathbf{V}_i + \frac{1}{N} E^\top E, \tag{12}$$

where $\frac{1}{N} B_i^2 \mathbf{V}_i^\top \mathbf{V}_i$ is the variance caused by regressor $\mathbf{V}_i$. Larger value indicates the regressor is more significant. Thus, the regressor $\mathbf{V}_i$ reduces the error $E$ at the following error reduction ratio $R_i$

$$R_i = \frac{B_i^2 \mathbf{V}_i^\top \mathbf{V}_i}{Y^\top Y}. \tag{13}$$

To select $n_s$ significant regressors, Gram-Schmidt orthogonalization method is adopted here.

For $i \in [1, n]$, we have

$$\mathbf{V}_1^{(i)} = \mathbf{X}_i,$$

$$B_1^{(i)} = \frac{\mathbf{V}_1^{(i)\top} Y}{\mathbf{V}_1^{(i)\top} \mathbf{V}_1^{(i)}},$$

$$R_1^{(i)} = \frac{\left(B_1^{(i)}\right)^2 \mathbf{V}_1^{(i)\top} \mathbf{V}_1^{(i)}}{Y^\top Y}, \tag{14}$$

Next, calculate the largest $R_1$ and represent as $R_1^{(i_{(1)})} = \max_i R_1^i$. The corresponding regressor is $\mathbf{V}_1 = \mathbf{V}_1^{(i_1)} = \mathbf{X}_{(i_1)}$.

For $i \in [1, n]$ at $k$th step and $i \neq i_1, \ldots, i_{k-1}$, we have

$$C_{jk}^{(i)} = \frac{\mathbf{V}_j^\top \mathbf{X}_i}{\mathbf{V}_j^\top \mathbf{V}_j} \quad \text{with} \quad j = 1, \ldots, k-1$$

$$\mathbf{V}_k^{(i)} = \mathbf{X}_i - \sum_{j=1}^{k-1} c_{jk}^{(i)} \mathbf{V}_j,$$

$$B_k^{(i)} = \frac{\mathbf{V}_k^{(i)\top} Y}{\mathbf{V}_k^{(i)\top} \mathbf{V}_k^{(i)}},$$

$$R_k^{(i)} = \frac{\left(B_k^{(i)}\right)^2 \mathbf{V}_k^{(i)\top} \mathbf{V}_k^{(i)}}{Y^\top Y}$$

(15)

Similarly, the maximum error reduction ratio is $R_k^{(i_k)} = \max_i R_k^{(i)}$, and the corresponding regressor is $\mathbf{V}_k = \mathbf{V}_k^{(i_k)} = \mathbf{X}_{i_k} - \sum_{j=1}^{k-1} C_{j_k}^{(i_j)} \mathbf{V}_j$.

If the value of unexplained output variance drops below the limit $\varepsilon$, that is, $1 - \sum_{j=1}^{n_s} R_j^{(i_j)} < \varepsilon$, the algorithm will be predetermined. Thus, the orthogonal basis vectors $\mathbf{V}_i$ are obtained. $\square$

