# OpenReview forum: "iQR: Quantile Regression with QR Orthogonal Decomposition for Resource Scheduling Optimization without Empirical Model"
_ICLR.cc/2025/Conference — ICLR 2025 Conference Withdrawn Submission_

### Official Review · Reviewer_Feka · 2024-10-25

**Soundness:** 3
**Presentation:** 3
**Contribution:** 3
**Rating:** 5
**Confidence:** 2

**Summary:**

A sparse system identification framework combining quantile regression and QR orthogonal decomposition is introduced in this paper to  solve resource scheduling optimization problems. The proposed iQR method is designed leverages Fourier Transform and orthogonal least squares to efficiently select basis vectors. This reduces computational complexity and enables precise, cost-effective scheduling without requiring extensive computational resources. Numerical experiments on multiple datasets demonstrate that iQR outperforms SOTA methods, achieving better scheduling accuracy while minimizing economic costs.

**Strengths:**

-The paper is well-organized in general.
-The numerical results are extensive and clear.

**Weaknesses:**

1. Could give more detailed description of the problem in the introduction
2. explain $\rho_{\tau}$ in eq. (3)
3. explain $\lambda$ in eq. (3) and how to select $\lambda$
4. need to formally define LASSO( ) before eq. (6) ; Also needs to explain (6) in general since many notations appears without definition.

**Questions:**

1. How do you select the fixed learning rate for all models in numerical experiment?
2. How do you solve problem 7?

---

### Official Review · Reviewer_6J47 · 2024-10-30

**Soundness:** 2
**Presentation:** 3
**Contribution:** 2
**Rating:** 3
**Confidence:** 3

**Summary:**

This paper presents iQR, a model-free framework for resource scheduling using quantile regression and QR decomposition to optimize demand coverage at minimal cost. iQR uses efficient basis selection to accelerate regression and prioritizes predictions that reliably meet resource needs. Tested on multiple datasets, iQR demonstrates superior scheduling performance over traditional neural networks, offering a fast, lightweight solution that operates without GPU support.

**Strengths:**

- This paper is clearly written.

- This paper introduces a new problem and designs a workflow to solve it.

**Weaknesses:**

- The iQR framework lacks sufficient novelty, as both the global and local identification components are adopted from [1].
- Lemma 1 is relatively trivial and does not add substantial value.
- The orthogonality condition in Theorem 1 is quite strong.
- The idea behind SOR+L1QR closely resembles sure independence screening in existing literature (see, e.g., [2-3]), but no discussion of this similarity is included.
- No theoretical guarantees are established for the iQR framework.

### Reference

- [1] Liu, Xiaoyi, et al. "Interpretable Sparse System Identification: Beyond Recent Deep Learning Techniques on Time-Series Prediction." The Twelfth International Conference on Learning Representations.

- [2] Fan, Jianqing, and Jinchi Lv. "Sure independence screening for ultrahigh dimensional feature space." Journal of the Royal Statistical Society Series B: Statistical Methodology 70.5 (2008): 849-911.

- [3] Sheng, Ying, and Qihua Wang. "Model-free feature screening for ultrahigh dimensional classification." Journal of Multivariate Analysis 178 (2020): 104618.

**Questions:**

- How does the framework handle non-stationary sequences?

- In equation (1), it appears that the indicator function is applied to a matrix? If so, this is somehow weird.

---

### Official Review · Reviewer_2C4U · 2024-11-04

**Soundness:** 2
**Presentation:** 2
**Contribution:** 2
**Rating:** 3
**Confidence:** 2

**Summary:**

This paper presents a quantile regression (QR) approach for time-series forecasting.   Standard approaches for time series prediction often do not account for practical requirements such as under/overestimation.  To address this, the authors propose iQR, which combines a basis selection strategy with quantile regression.  Empirically, they demonstrate that their approach achieves results are competitive in terms of minimizing QRE and positive mean absolute/squared error.

**Strengths:**

- In the numerical experiments, the iQR approach consistently achieves the best results in minimizing QRE and positive mean absolute/squared error.
- Their approach is made efficient through the use of decomposition and stepwise regression

**Weaknesses:**

- **Clarity and Motivation.**  Overall, the paper suffers from a lack of clarity around critical aspects of the paper.  For example, the term business requirements/constraints is mentioned dozens of times in the first couple of sections of the paper without any formal definition being provided.  This makes the motivation and use case for this work very hard to contextualize.  In addition, it is unclear how useful their definition of business constraints.  For example, their approach would not be able to address more complex constraints or resource scheduling problems that can utilize overproduction from the previous periods in the next period.  As such, I believe it is essential to define and discuss the limitations of their problem settings and methodology appropriately to improve the clarity and motivation of the paper.
- **Experiments.** The numerical experiments give rise to several weaknesses.     In general, the below points lead me to believe that more robust evaluation and fair comparisons are needed to strengthen the contribution.
   - Almost all of the benchmarks are from standard time series applications, and the need or suitability for evaluating with respect to quantile regression error (QRE) is not clear.
   -  All comparisons against other methods are evaluated using QRE and PMSE/PMAE.  These metrics will undoubtedly favor iQR as this is the only method optimized to perform well in this setting.
   - Moreover, important baselines, i.e., any mentioned in the section on \textbf{Quantile Regression-based scheduling}, e.g., FLM, are not compared against.  These would certainly provide a more fair evaluation.

**Questions:**

- Are there other metrics that may be useful in evaluating iQR more robustly?
- Are there any datasets that specifically highlight the business constraints that the authors focus on?
- Why were other QR baselines not evaluated?

Minor Remark:  Please double-check the spelling/grammar. I found a lot of instances wherein I needed to reread sentences due to a lack of clarity.

---

### Official Review · Reviewer_eHyY · 2024-11-05

**Soundness:** 3
**Presentation:** 2
**Contribution:** 2
**Rating:** 5
**Confidence:** 3

**Summary:**

The authors proposed an efficient quantile regression approach for resource scheduling problems. The proposed framework processes the time series in two subsequent phases, global and local information separately. In both phases, the different basis vectors are derived using Fast Fourier Transformation (FFT) and then stepwise orthogonal regression (SOR) is employed to choose the most important basis. Finally the selected basis vectors are passed to Quantile Regression with L1 norm to predict the output series. Additionally in local identification phase, the basis vectors are decomposed with seasonal trend decomposition (STL) before passing to quantile regression for local prediction.

**Strengths:**

1. Unlike Neural Network based models that often overlook the time-series nature of the input series, the authors utilized time series specific features such as basis vectors from FFT and season/trend from STL to predict the allocated resource, which result in accurate predictions.

2. The main advantage of the approach is its robustness.  Training regression with lasso term is computationally less costly than the NN approaches such as LSTM and Transformers based models.

**Weaknesses:**

1. My main concern about the study is that all baselines are for time series prediction and not meant for resource scheduling optimization. As the baselines' objective is not to exceed the ground truth, it is not surprising the proposed method outperforms the baselines.

2. Another issue about the baselines is, while some Dlinear and FEDformer use time series based features, all the baselines are NN and transformer based. In the literature review, the authors acknowledge that the unsatisfactory performances of transformer based methods in some cases. It is important to have QR based methods in the study to for fair comparison.

**Questions:**

See the weaknesses.

---

### Note · Authors · 2024-11-21

I have read and agree with the venue's withdrawal policy on behalf of myself and my co-authors.